# Relating multivariate shapes to genescapes using phenotype-biological process associations for craniofacial shape

Jose D Aponte[1], David C Katz[1], Daniela M Roth[2], Marta Vidal-García[1], Wei Liu[1], Fernando Andrade[3], Charles C Roseman[3], Steven A Murray[4], James Cheverud[3], Daniel Graf[2,5], Ralph S Marcucio[6]*, Benedikt Hallgrímsson[1,7]*

[1]Department of Cell Biology & Anatomy, Alberta Children's Hospital Research Institute and McCaig Bone and Joint Institute, Cumming School of Medicine, University of Calgary, Calgary, Canada; [2]School of Dentistry, Faculty of Medicine and Dentistry, University of Alberta, Edmonton, Canada; [3]Department of Biology, Loyola University Chicago, Chicago, United States; [4]The Jackson Laboratory, Bar Harbor, United States; [5]Department of Medical Genetics, Faculty of Medicine and Dentistry, University of Alberta, Edmonton, Canada; [6]Department of Orthopaedic Surgery, School of Medicine, University of California, San Francisco, San Francisco, United States; [7]Department of Animal Biology, University of Illinois Urbana Champaign, Urbana, United States

*For correspondence:
ralph.marcucio@ucsf.edu (RSM);
bhallgri@ucalgary.ca (BH)

**Abstract** Realistic mappings of genes to morphology are inherently multivariate on both sides of the equation. The importance of coordinated gene effects on morphological phenotypes is clear from the intertwining of gene actions in signaling pathways, gene regulatory networks, and developmental processes underlying the development of shape and size. Yet, current approaches tend to focus on identifying and localizing the effects of individual genes and rarely leverage the information content of high-dimensional phenotypes. Here, we explicitly model the joint effects of biologically coherent collections of genes on a multivariate trait – craniofacial shape – in a sample of n = 1145 mice from the Diversity Outbred (DO) experimental line. We use biological process Gene Ontology (GO) annotations to select skeletal and facial development gene sets and solve for the axis of shape variation that maximally covaries with gene set marker variation. We use our process-centered, multivariate genotype-phenotype (process MGP) approach to determine the overall contributions to craniofacial variation of genes involved in relevant processes and how variation in different processes corresponds to multivariate axes of shape variation. Further, we compare the directions of effect in phenotype space of mutations to the primary axis of shape variation associated with broader pathways within which they are thought to function. Finally, we leverage the relationship between mutational and pathway-level effects to predict phenotypic effects beyond craniofacial shape in specific mutants. We also introduce an online application that provides users the means to customize their own process-centered craniofacial shape analyses in the DO. The process-centered approach is generally applicable to any continuously varying phenotype and thus has wide-reaching implications for complex trait genetics.

## Editor's evaluation

This paper offers a new take on multivariate genotype-phenotype mapping that identifies the joint phenotypic effect of genes involved in known biological processes that impact craniofacial variation. More specifically, the work expands on the traditional idea of candidate gene investigations into candidate biological process investigations, grouping multiple genes into a single analysis. In doing so, the authors show the joint effects of three strong candidate processes, chondrocyte differentiation, determination of left/right symmetry, and palate development on multidimensional craniofacial shape in the heterogenous Diversity Outbred mouse population.

## Introduction

Variation in human craniofacial shape is moderately to highly heritable (~30–70%; *Cole et al., 2017*; *Tsagkrasoulis et al., 2017*), and resemblances among close relatives as well as twins underscore the strong relationship between shared genetics and shared phenotype (*Johannsdottir et al., 2005*; *Nakata, 2014*). Despite many studies in humans and in mice (*Claes et al., 2018*; *Cole et al., 2016*; *Shaffer et al., 2016*), however, we know very little about the genetic basis for variation in craniofacial shape. This is likely due to genetic complexity (*Katz et al., 2019*; *Richtsmeier and Flaherty, 2013*; *Visscher, 2008*; *Wood et al., 2014*; *Wray et al., 2013*). Like many aspects of morphological variation, craniofacial shape is extraordinarily polygenic. Genes with major mechanistic roles in facial development such as *Fgf8* often contribute little to observed phenotypic variation (*Green et al., 2017*) while genetic influences without obvious connections to craniofacial development emerge as significant contributors (*Kenney-Hunt et al., 2008*; *Klingenberg and Leamy, 2001*; *Maga et al., 2015*; *Pallares et al., 2015*; *Pallares et al., 2014*). The effects of genetic variants on phenotype often depend on genetic background (*Mackay and Moore, 2014*; *Percival et al., 2017*), and many mutations have variably penetrant effects even when background is controlled (*Kawauchi et al., 2009*; *Rendel, 1967*). These issues likely arise because genetic influences act through multiple layers of interacting developmental processes to influence phenotypic traits, resulting in complex patterns of epistasis and variance heterogeneity (*Hallgrimsson et al., 2019*; *Hallgrimsson et al., 2014*; *Kawauchi et al., 2009*; *Wagner and Zhang, 2011*; *Gasch et al., 2016*). Solutions that go beyond studies of single gene effects are required to overcome these significant challenges in complex trait genetics. Here, we implement an enhanced form of the more general candidate gene approach to evaluate the conjoint effects of multiple genes on a complex trait – craniofacial shape.

There are two basic approaches to mapping genetic effects on to phenotypic variation. A candidate gene approach measures genotypic values with known physiological and biochemical relationships to the phenotypes of interest (*Cheverud and Routman, 1996*). In contrast, a random marker or genome-wide approach seeks to associate any potential genetic variant with variation in the trait of interest. There are advantages and disadvantages to these two approaches. The candidate gene approach is blind to the unknown – phenotypic variation is often associated with genes not expected to be important. On the other hand, a candidate gene approach allows direct measurement of genotypic values and produces results that are interpretable in terms of trait physiology or development. A genome-wide or random marker approach can produce unexpected insight by revealing novel gene-phenotype associations. However, this comes at a great cost in power (*Visscher et al., 2017*). For highly polygenic traits, this approach often produces a 'tip of the iceberg' effect in which studies reveal a small and often incoherent subset of the genes that actually determine variation in the trait of interest (*Broman, 2009*).

Several strategies have been developed that partially overcome these tradeoffs. One solution is the use of polygenic risk scores. Polygenic risk scores assess the overall genetic influence on a trait without regard to the genome-wide significance of individual SNP effects (*Dudbridge, 2013*; *Wray et al., 2007*). Approaches such as meta-analyses of genome-wide association studies (GWAS) or studies based on extreme phenotypes *Morozova et al., 2015* have expanded gene lists for a variety of complex traits. However, lengthy lists of genes or overall genomic risk for specific phenotypes do not necessarily constitute tractable genetic explanations for phenotypic variation. When thousands of genes are required to explain heritable variation in stature, for instance, it is not clear what such lists tell you beyond the obvious fact that stature is heritable and polygenic (*Yang et al., 2010*; *Wood et al., 2014*). This tension between hypothesis-driven and hypothesis-free approaches and their attendant tradeoffs between statistical power and interpretability is, arguably, a major issue within complex

trait genetics. To resolve this conceptual conflict, approaches are needed that integrate quantitative genetics with biological insights regarding the cellular and developmental processes through which genes influence phenotypic variation.

Existing approaches to complex trait genetics also tend to treat phenotypic traits as singular and one-dimensional. Even for morphological variation, most studies reduce shape variation to linear distances, principal components (PCs), regression scores or measures of size that are then mapped as individual traits (*Xiong et al., 2019*; *Shaffer et al., 2016*; *Cole et al., 2016*). This approach disregards the information content of multivariate phenotypic variation. While univariate traits only vary along one dimension, high-dimensional traits such as craniofacial shape can vary in direction as well as magnitude within a multidimensional shape space. To identify the distinctive axes of gene effects on a multivariate trait, one must model such multiple multivariate relationships directly.

Building on Mitteroecker et al.'s (2016) multivariate genotype-phenotype (MGP) method, we extend the candidate gene framework to evaluate the combined contributions of genes to variation in high-dimensional phenotypic traits such as craniofacial shape. Grouping genes by ontological information such as membership in developmental pathways or other relevant biological hypotheses, our process-centered multivariate approach, herein referred to as process MGP, brings traditional GWAS together with a simplified model of the hierarchical genotype-phenotype (GP) map. Gene Ontology (GO) terms are broadly grouped into three categories:cellular components, molecular functions, and biological processes. Our work focuses on biological process gene annotations because they group known relationships between several genes that contribute to a developmental function. The process MGP approach aims to leverage this knowledge by modeling the joint effects of these genes on craniofacial shape variation.

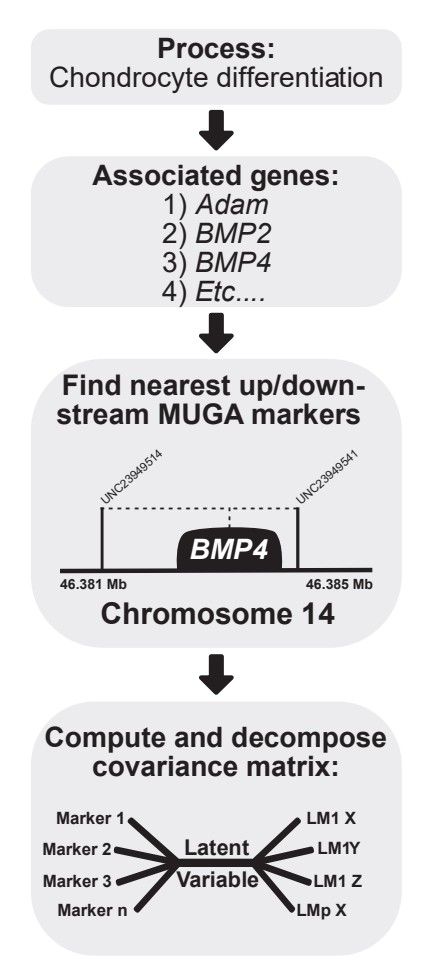

**Figure 1.** Process multivariate genotype-phenotype (MGP) schematic. Once a process is selected, we cross-reference the known gene locations using Ensembl with the locations of the genotyped markers in the Diversity Outbred (DO) sample. The founder probabilities of the nearest upstream and downstream markers are averaged for each gene. The compiled founder probabilities and landmark coordinates are then used in a regularized partial least squares (PLS) model to estimate the axis of greatest covariance between the marker data and craniofacial variation.

Understanding the genetic determinants of craniofacial variation, as with most complex traits, represents a many-to-many GP map problem (*Lewontin, 1974*; *Houle et al., 2010*). Both phenotypic and genotypic measurements have complex within-set covariance structures. On the genetic side, the covariance structure is represented by pathway/biochemical interactions, as well as chromosomal structure like linkage, chromatin, and 3D chromosomal organization. For shape-related phenotypes, the covariance matrix is structured by the chosen set of landmarks and their resulting coordinates. The functional relationship from genotype to phenotype is then described by a between-set covariance (*Klingenberg and Leamy, 2001*; *Mitteroecker et al., 2016*). To dissect these relationships, we use a regularized partial least squares (PLS) (*Lorenzo et al., 2019*) approach to estimate a low-dimensional mapping from the alleles in our sample to variation in adult mouse craniofacial shape. While PLS is well suited for analysis of covariation between two sets of measurements, regularization is essential for

mitigating overfitting when there are many alleles simultaneously modeled. We focus on how allelic variation in processes relevant to craniofacial development maps to craniofacial shape variation. We ask the following four questions:

1. How much shape variation is communally accounted for by genes contributing to a process, for example, chondrocyte differentiation?
2. How similar are the effects of different processes on shape? For instance, do cell proliferation genes affect face shape in a similar way to genes in the bone morphogenetic protein pathway?
3. How similar are mutant model effects and process effects? For example, do chondrocyte mutant effects align with the effects of natural variants in chondrocyte differentiation genes?
4. Can one use the similarity of a mutational effect to MGP process effects predict unobserved phenotypes associated with that mutation?

Together, these questions demonstrate the ability of the process MGP approach to add meaningful understanding of the complex relationships between genotype and phenotype by quantifying higher-level regularities between complex phenotypic and genomic data. We also demonstrate its potential as a resource for the study of mutational effects on complex traits such as craniofacial shape.

## Results
### Process MGP mapping

For process MGP analyses, we used the mouse genome informatics database (*Bult et al., 2018Kanehisa et al., 2017*) to identify genes annotated to a given process. Each annotation term has an associated GO ID. For example, 'chondrocyte differentiation' has GO ID GO:000206 (*Figure 1*, box 1). We cross-reference the GO ID with the Ensemble genome database (GRCm38.p6) to find the name, chromosome, and base pair start/end position for each gene (*Figure 1*, box 2) annotated to the process. For genes with multiple splice variants, we select the longest transcript. For each gene, we compare marker base pair positions and select the closest upstream and downstream markers to the center of each gene. The eight-state genotype probability is then calculated as the average founder allele probabilities between the two selected markers. (*Figure 1*, box 3). After marker selection, we fit a regularized PLS model using the founder allele probabilities (eight variables/marker) and full landmark data set (*Figure 1*, box 4). Regularization penalizes the coefficients such that increasing regularization strength causes more coefficients to have a value of zero. We chose a regularization parameter using 10-fold cross-validation. For each of the example process MGP analyses shown, we chose the regularization strength that best represented the tradeoff between minimizing model error and maximizing interpretability of marker effects and the similarity of phenotypic effects with mouse mutant models. The full cross-validation results are shown in *Figure 2*, Figure 4, and Figure 5—figure supplement 1.

We demonstrate process MGP mapping with three examples. The first estimates the primary axis of skull shape covariation with genes annotated to 'chondrocyte differentiation' (*Figure 2*). Differentiation of chondrocytes is one of several key developmental processes involved in endochondral ossification. Endochondral bones form the majority of the cranial base through a cartilage model of bone formation (*Percival and Richtsmeier, 2013*). There are 38 genes annotated to chondrocyte differentiation in the Ensembl database (*Yates et al., 2020*). In the figure, genetic effects are shown as zero-centered bars that span the range of estimated allele effects across the eight DO founders; individual founder allele effects – eight per marker – are color-coded within those bars (*Figure 2A*). We chose a regularization parameter of 0.075 for this analysis (*Figure 2—figure supplement 1*). Among chondrocyte differentiation genes, *Nov/Ccn3*, *Bmpr1b* (*Alk6*), and *Nfib* are most implicated in the major axis of pathway covariation with craniofacial shape. The phenotypic effects at each landmark primarily relate to anteroposterior positioning of the zygomatic arches and dorsoventral jugal position (*Figure 2B*). The chondrocyte differentiation GP map explains 2.15% of the total variance in craniofacial shape. Compared to 10,000 random permutations of the model, chondrocyte differentiation explains substantially more craniofacial variation (*Figure 2—figure supplement 2*).

*Figure 2C* compares the direction of the chondrocyte differentiation MGP axis – magnified 4× – to the axis of shape variation of a relevant mutant phenotype. We chose homozygous *Bmpr1b* mutants for this comparison for three reasons. The first is because Bmpr1b in synergy with other bone morphogenic protein pathway receptors regulates chondrocyte proliferation and differentiation in embryonic cartilage condensations (*Yoon et al., 2005*, ). The second reason we chose *Bmpr1b*

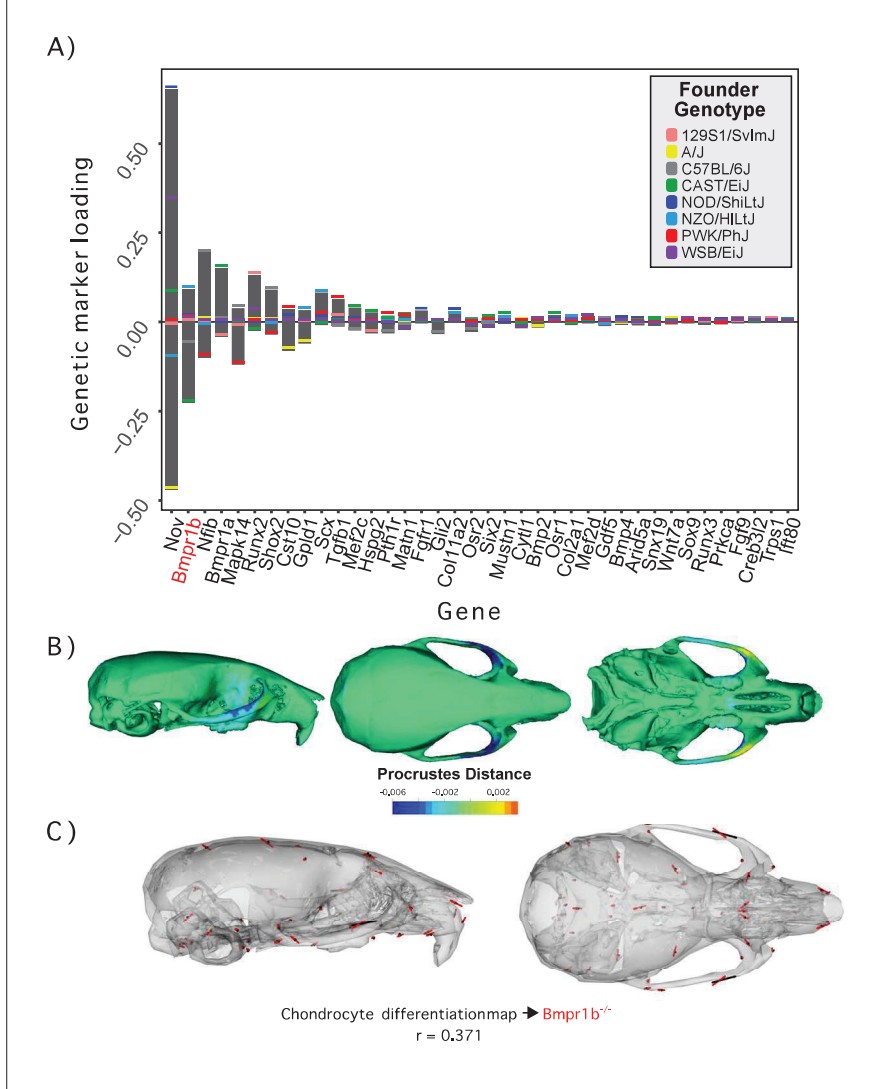

**Figure 2.** Process multivariate genotype-phenotype (MGP) for chondrocyte differentiation with a regularization parameter of 0.075. (**A**) PLS1 genetic loadings are shown for each gene in the model sorted from largest to smallest effects. Individual founder allele effect sizes are colored within each bar. The gene in red text corresponds to the mutant used for comparison of phenotypic effects. (**B**) The estimated chondrocyte differentiation MGP phenotype is shown with a heatmap. Warm colors represent areas of relative expansion, light green represents areas of little shape effect, and cool colors represent areas with relative contraction. (**C**) Chondrocyte differentiation MGP effects shown in black vectors multiplied 4× are compared to a *Bmpr1b* (*Alk6*) homozygous mutant and are shown with red vectors. The vector correlation between chondrocyte differentiation MGP and *Bmpr1b* is shown below the phenotypic effects.

The online version of this article includes the following figure supplement(s) for figure 2:

**Figure supplement 1.** 10-fold cross-validation results for the chondrocyte differentiation multivariate genotype-phenotype (MGP).

**Figure supplement 2.** Permutation of marker sets of fixed size.

mutant comparisons is because the marker selected for *Bmp1rb* in the genomic analysis contains one of the strongest allelic effects associated with the morphological effect. *Bmpr1b* shows stronger loading/association than *Bmpr1a* (*Figure 2A*). While *Bmpr1a* has well-established roles in craniofacial development (*Liu et al., 2005*; *Liu et al., 2018*), the role for *Bmpr1b* on its own is less clear. *Bmpr1b* mutants show shorter long bones at birth and overexpression of a dominant negative *Bmpr1b* using a type 1 collagen promoter showed delayed ossification of the frontal, parietal, and occipital bones

(*Yoon et al., 2005*; *Zhao et al., 2002*). The overall phenotypic directions of *Bmpr1b* mutant variation and chondrocyte differentiation variation are moderately correlated at $r$ = 0.371 (t = –5.06, df = 160, p<0.0001), but the direction at landmarks with large effects in mutant and MGP is clearly coincident. Over the landmarks we measured, the chondrocyte differentiation effect is less global than the *Bmpr1b* effect, likely due to the difference in severity of the mutant phenotype.

The similarity of the chondrocyte differentiation effect with the *Bmpr1b* mutant and the high loading *Bmpr1b* allele in the DO genome suggest that *Bmpr1b* mutants may produce chondrocyte differentiation defects in the developing neurocranium. In response to the process MGP results, quantified cell size and distribution in the intersphenoid synchondroses (ISS) of several mutant and control *Bmpr1b* mice (*Figure 3—figure supplement 1*). We found that homozygotes show overall larger cell sizes as well as a differing distribution of cell sizes throughout the width of the ISS (*Figure 3A–C*; $\chi^2$ = 21.23, df = 3, p<0.0001). The presence of larger cell sizes in the homozygote *Bmpr1b* mutants suggests that the synchondroses possess more hypertrophic chondrocytes. Additionally, *Bmpr1b* homozygous mutant mice show premature fusion of the coronal suture (*Figure 3D*). Both features have not been reported in the literature.

The second example quantifies cranial shape covariation with the 81 genes annotated to 'determination of left/right symmetry.' We used a regularization parameter of 0.04 (*Figure 4—figure supplement 1*). There are several high loading alleles that contribute to the determination of left/right symmetry MGP phenotype. In particular, an *Fgf10* allele inherited from the Castaneus founder background was among the most important (*Figure 4A*). FGF10 is a key ligand in early development, directing proliferation as well as differentiation for many craniofacial components, including the palate, teeth, and bones (*Hilliard et al., 2005*; *Prochazkova et al., 2018*; *Watson and Francavilla, 2018*). The phenotype associated with left/right symmetry alleles is predominately related to a larger neurocranium volume relative to the outgrowth of the face (*Figure 4B*). We also visualized the asymmetry in the phenotypic response, which shows subtle asymmetry, particularly in the position of the anterior zygomatic landmark (*Figure 4D*). Several Fgf ligands including *Fgf10* are integral in the asymmetric distribution of organs (*Hecksher-Sørensen et al., 2004*). *Fgf8* has also previously been shown to produce asymmetric craniofacial phenotypes in zebrafish, but craniofacial asymmetry has not previously been observed in *Fgf10* mutants (*Albertson and Yelick, 2005*). Left/right symmetry loci explain 3.4% of the total variance in craniofacial shape, which exceeds the variance explained by 10,000 randomly permuted L/R symmetry MGP analyses (*Figure 4—figure supplement 2*). We compared the estimated L/R symmetry MGP effect with the direction of an *Fgf10* homozygous mutant because of the relative importance of the allelic effect (*Figure 4C*). The vector correlation between the *Fgf10* mutant and the estimated left/right symmetry effect is 0.672 (t = 12.29, df = 160, p<0.0001). The importance of other Fgf ligands in craniofacial symmetry, as well as the high-loading *Fgf10* allele for left/right symmetry MGP along with the similar genomic and mutant phenotypes, suggests that *Fgf10* mutants could show directional asymmetry in the cranium. To test this hypothesis, we measured a sample of 8 *Fgf10* adult mutant crania for object symmetry and detected significant directional asymmetry (*Figure 4D*; $F$ = 4.91, df = 52, p<0.0001).

The final example estimates the shape covariation attributed to the 73 genes annotated to 'palate development.' Formation and fusion of the palatal shelves are crucial for proper orofacial development and heavily influence overall facial shape (*Greene and Pisano, 2010*). We used a regularization penalty of 0.05 because it best balances the vector correlation to the mutant comparison and the reduction of prediction error (*Figure 5—figure supplement 1*). Several genes contribute strongly to the palate development MGP effect including *Ephb2, Gli3,* and *Lrp6* (*Figure 5A*). The estimated phenotype shows corresponding variation in palate length as well as strong effects in the majority of the cranial base landmarks (*Figure 5B*). Palate development MGP loci explain 2.4% of the total variance in cranial shape, which is greater than the variance explained by 10,000 randomly permuted palate development MGP models (*Figure 5—figure supplement 2*). We compared the palate development phenotype to a heterozygous *Ankrd11*, neural crest-specific knockout mouse. The *Ankrd11* locus is associated with KBG syndrome in humans, which presents with generally delayed bone mineralization as well as craniofacial characteristics including palate abnormalities (*Low et al., 2016*). While the vector correlation between the palate development MGP effect and the *Ankrd11* mutant over the complete set of cranial landmarks is moderate at $r$ = 0.339 (*Figure 5C*; t = 4.31, df = 160, p=0.0001), the vector correlation for palate landmarks is substantially higher at $r$ = 0.536.

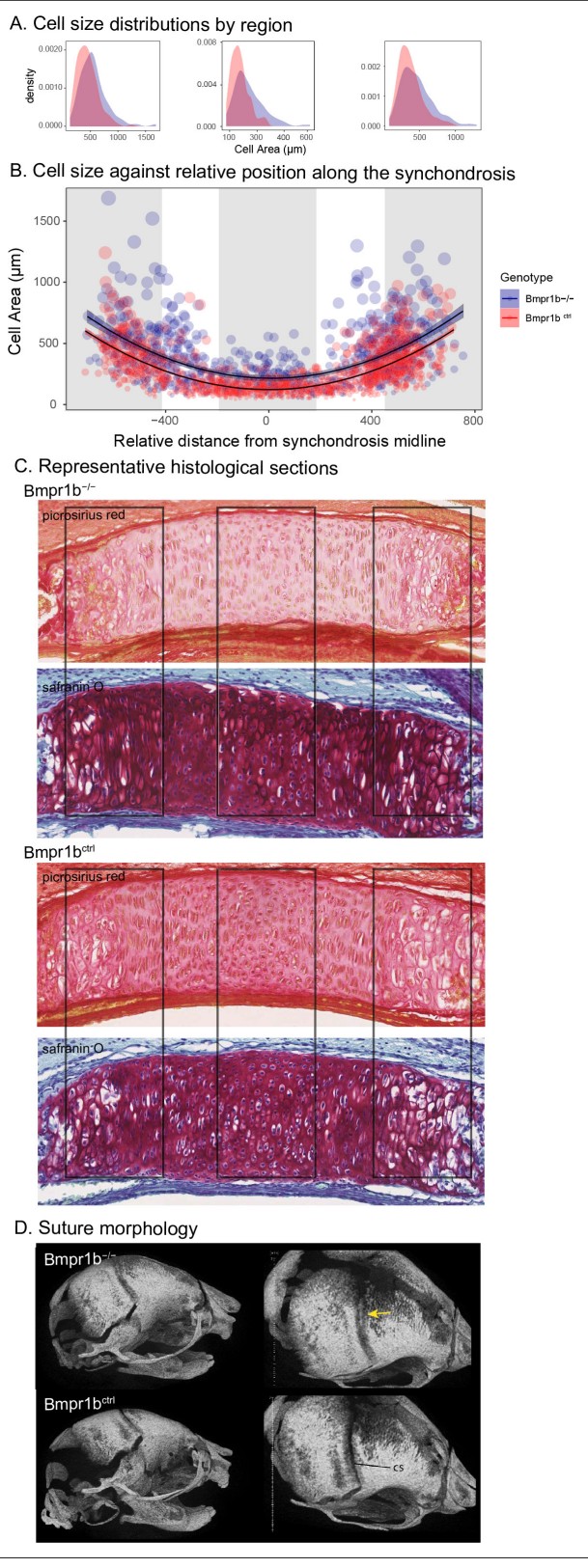

**Figure 3.** Chondrocyte defects in *Bmpr1b* mutants. (**A, B**) Quantification of cell size in the sections of the intersphenoid synchondrosis shows an increase in relative cell size as well as a change in the distribution of cell sizes throughout the width of the synchondrosis. (**C**) Sections of intersphenoid synchondroses highlighting the midline and extremes of the synchondroses. (**D**) Premature fusion of the coronal suture is visible in *Bmpr1b*

*Figure 3 continued on next page*

*Figure 3 continued*

homozygous mutants.

The online version of this article includes the following figure supplement(s) for figure 3:

**Figure supplement 1.** Chondrocyte morphometric example.

In each case above, we have shown how association of gene sets and phenotypic variation can produce highly informative results that can guide subsequent hypothesis testing. For a given biological process, we identified several genes that load strongly on the primary axis of MGP covariation for which mutant samples were available to us, as well. Future investigations could also use this information about genes with high loadings to generate new mutants for analysis of associated developmental processes. For each example, we focus only on the first PLS axis, so distinct joint gene effect combinations may contribute to novel phenotypic directions in lower PLS axes.

## Joint versus single-loci effects

While the process MGP approach focuses on the joint effect of markers on craniofacial shape, it is important to measure the extent that joint effects matter for craniofacial shape. Unlike alternative models such as canonical correlation analysis (CCA), the PLS model used for process MGP does not allow for statistical tests of individual marker effects. However, it is possible to measure the similarity of phenotypic effects after successively removing the most heavily loading markers from the full model. *Figure 6* shows the change in variance explained (A) as well as the change in the direction of phenotypic response (B) as markers are increasingly removed from the model. For each example analysis, we remove markers in order from most heavily loaded to least heavily loaded. We found that process MGP analyses with few loci of very large effect, like *Ccn3/Nov* for chondrocyte differentiation MGP, are very sensitive to the removal of the most highly loaded genes. The vector correlation between the full chondrocyte differentiation MGP model and the model with the *Ccn3/Nov* marker removed is 0.346.

For process MGP analyses with a more uniform distribution of marker effects, we found that the phenotypic effect is much more reliant on a multitude of marker alleles. For instance, L/R symmetry MGP with the 10 most heavily loaded markers removed still produced a vector correlation of 0.95 with the full model. The majority of process MGP analyses demonstrate a similar importance to several alleles, highlighting the main strength of process-level analyses over individual marker tests (*Figure 6—figure supplement 1*). In the following sections, we will examine how process MGP phenotypes relate to each other, as well as the phenotypic directions of several mutant mouse models.

## Pairwise comparison of craniofacial development processes

We chose 15 processes integral to craniofacial development and compared the pairwise similarity of effect on craniofacial shape using a heatmap based on clustering of the correlation matrix (*R Development Core Team, 2017*). Processes with similar effects on craniofacial shape will be highly correlated, while processes that affect distinct aspects of craniofacial variation will be uncorrelated to each other. The clustering algorithm resulted in two main blocks of strongly correlated effects (*Figure 7A*). The largest block of highly correlated phenotypic effects includes neural crest cell migration, epithelial to mesenchymal transition, forebrain development, as well as some of the most general developmental processes like cell proliferation, bone development, apoptosis, A/P pattern specification, and FGFR signaling. In addition, there is a general BMP block, with Bmp signaling, dorsoventral pattern formation, endochondral ossification, and positive regulation of skeletal muscle tissue growth. Interestingly, phenotypic variation associated with cranial suture morphogenesis, neural tube patterning, and intramembranous ossification is largely uncorrelated with the other craniofacial developmental processes included here.

To assess the stability of the clustering result, we estimated the vector correlation between the cluster distances – also known as the cophenetic distance – and the original correlation matrix (*Sneath and Sokal, 1973*). A high vector correlation suggests reliable clustering, whereas a low correlation suggests a random clustering result. The correlation between the cophenetic distance matrix and the correlation matrix is 0.648 (t = 8.64, df = 103, p=$7.6 \times 10^{-14}$), suggesting a moderate, though significant, structure in the similarity of effects amongst this set of MGP processes.

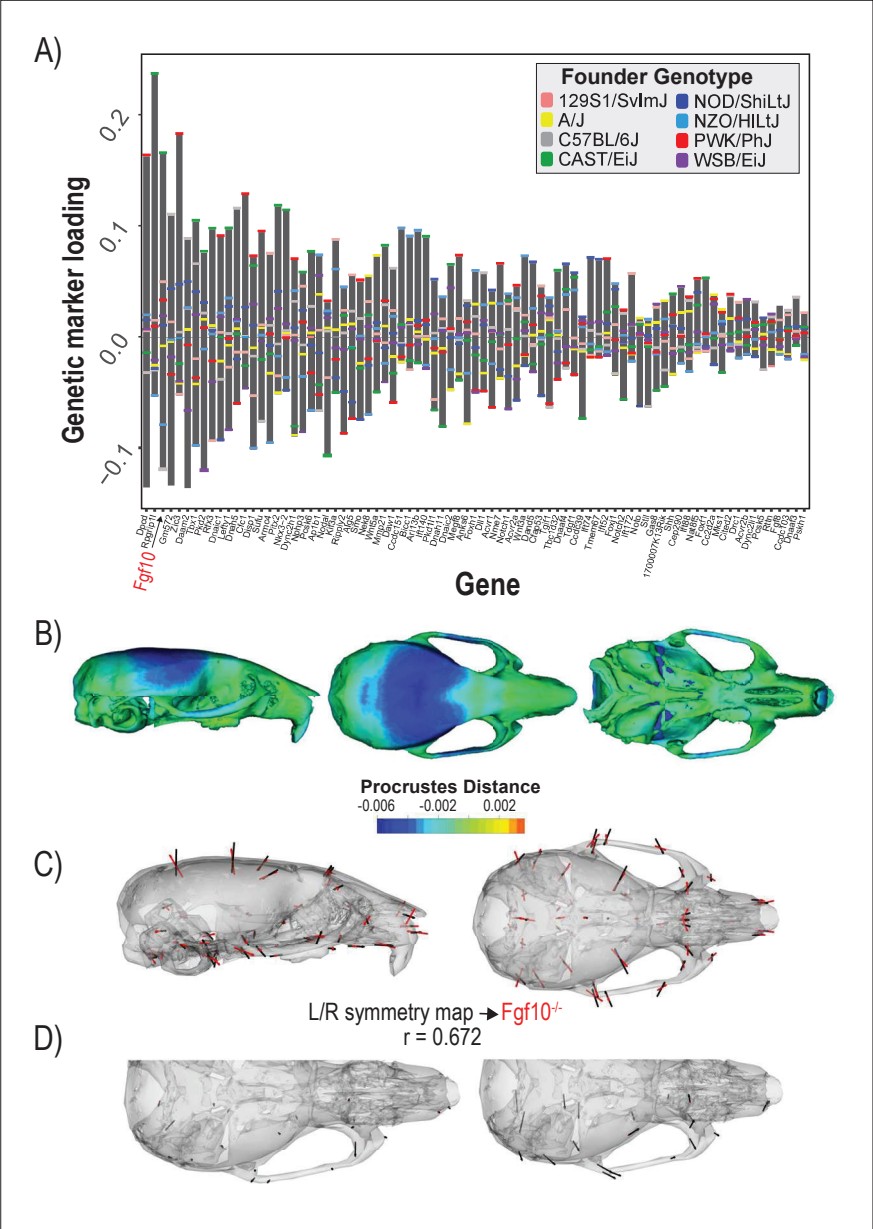

**Figure 4.** Process multivariate genotype-phenotype (MGP) for determination of left/right symmetry with a regularization parameter of 0.04. (**A**) PLS1 genetic loadings are shown for each gene in the model sorted from largest to smallest effects. Individual founder allele effect sizes are colored within each bar. The gene in red text corresponds to the mutant used for comparison of phenotypic effects. (**B**) The estimated left/right symmetry MGP phenotype is shown with a heatmap. Warm colors represent areas of relative expansion, light green represents areas of little shape effect, and cool colors represent areas with relative contraction. (**C**) Estimated left/right symmetry MGP phenotype is shown with black vectors multiplied 4×. An *Fgf10* homozygous mutant is shown with red vectors for comparison. The vector correlation between left/right symmetry MGP and the *Fgf10* mutant is shown below the phenotypic effects. (**D**) Visualizations of asymmetry in the L/R MGP response and the *Fgf10* homozygous mutant. Asymmetry vectors are magnified 4×.

The online version of this article includes the following figure supplement(s) for figure 4:

**Figure supplement 1.** 10-fold cross-validation results for the chondrocyte differentiation multivariate genotype-phenotype (MGP).

**Figure supplement 2.** The permuted R² distribution of 10,000 L/R symmetry multivariate genotype-phenotype (MGP) analyses is shown in blue.

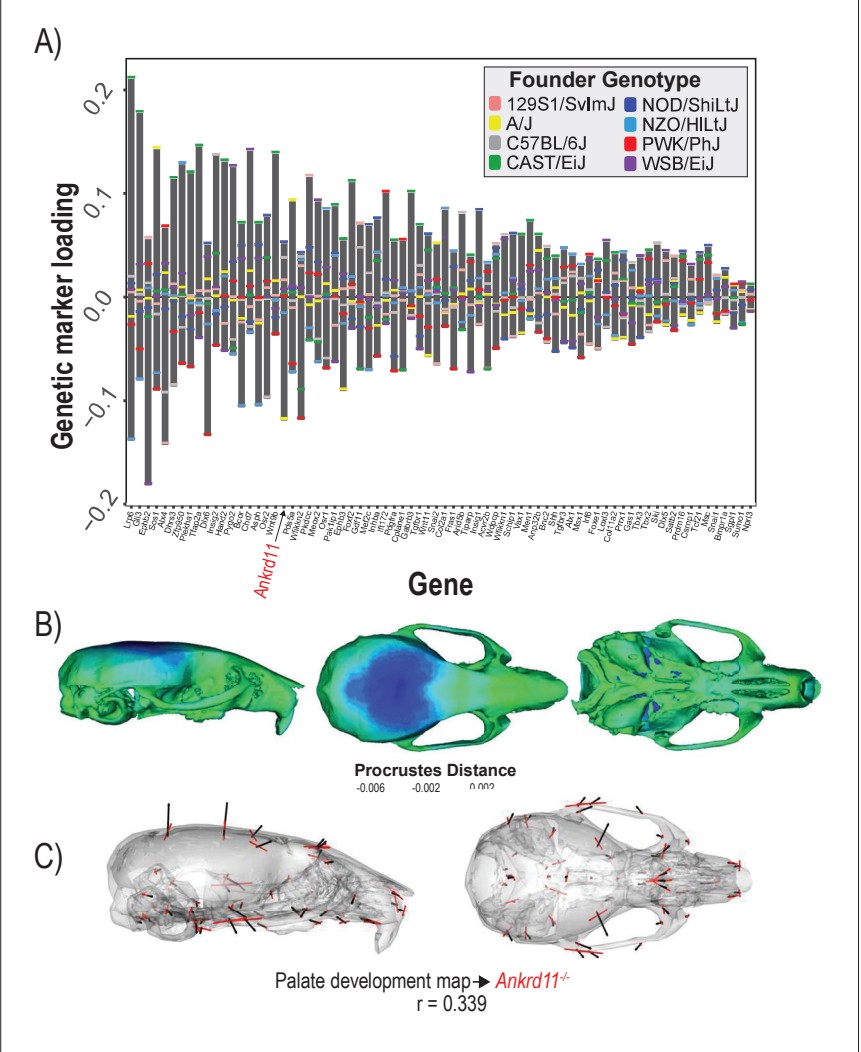

**Figure 5.** Process multivariate genotype-phenotype (MGP) for palate development. (**A**) PLS1 genetic loadings are shown for each gene in the model sorted from largest to smallest effects. Individual founder allele effect sizes are colored within each bar. The gene in red text corresponds to the mutant used for comparison of phenotypic effects. (**B**) The estimated palate development MGP phenotype is shown with a heatmap. Warm colors represent areas of relative expansion, light green represents areas of little shape effect, and cool colors represent areas with relative contraction. (**C**) Estimated palate development MGP phenotype is shown with black vectors multiplied 4×. An *Ankrd11* mutant mean is shown with red vectors for comparison. The vector correlation between palate development MGP and the *Ankrd11* mutant is shown below the phenotypic effects.

The online version of this article includes the following figure supplement(s) for figure 5:

**Figure supplement 1.** 10-fold cross-validation results for the palate development multivariate genotype-phenotype (MGP).

**Figure supplement 2.** The permuted R$^2$ distribution of 10,000 palate development multivariate genotype-phenotype (MGP) analyses is shown in blue.

The similarity in process MGP effects suggests that processes may coordinate in a limited set of potential directions of phenotypic variation. One reason that we could observe this pattern that is not because of common axes of GP variation is that key genes show up repeatedly within processes and largely drive these patterns of phenotypic variation. *Figure 7B* shows over 45,000 pairwise process MGP vector correlations as a function of the number of shared genes between the two randomly chosen annotations. While the similarity of phenotypic effects generally increases as the number of shared genes increases, GO processes that share genes are not necessarily strongly correlated. For GO processes that share between 0 and 10 genes, the observed correlations in phenotypic response

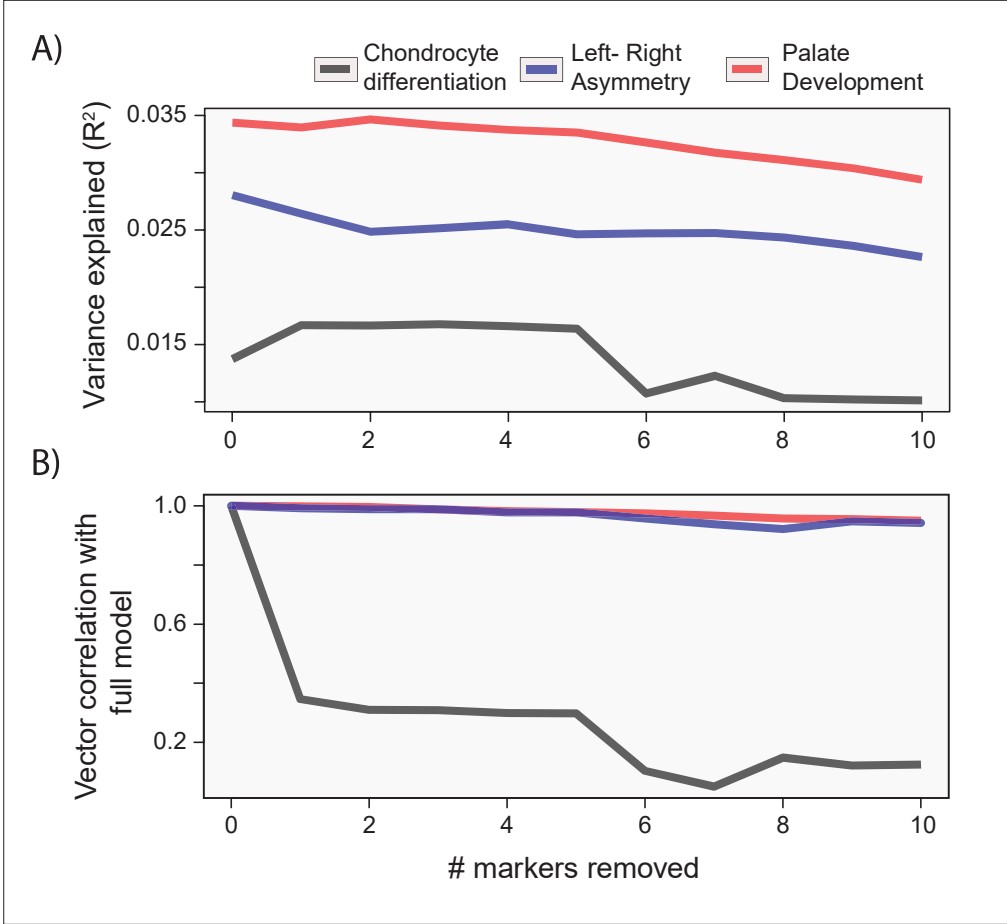

**Figure 6.** Gene drop tests. For each of the example analyses, we show the effect of removing the most heavily loaded markers from the process multivariate genotype-phenotype (MGP) analysis on the (**A**) variance explained by the model and (**B**) vector correlation with the full model. The variance explained as well as vector correlation is relatively stable for both L/R symmetry and palate development MGP models, suggesting that the effect is driven by the coordination of many markers. In contrast, chondrocyte differentiation MGP shows large differences, particularly in the direction of the phenotypic effect as the most heavily loaded markers are removed from the analysis.

The online version of this article includes the following figure supplement(s) for figure 6:

**Figure supplement 1.** Single marker importance.

spanned from no correlation to almost entirely concordant. GO processes that share more than 10 genes show generally higher vector correlations, with the lowest vector correlation we observed at 0.35.

## Process effects in the mutant morphospace

To assess the extent to which craniofacial shape variation associated with developmental processes aligns with variation from mutants of major effect, we projected seven process effects onto the first two PCs of a dataset containing the DO sample, and samples from 30 mutant genotypes (*Figure 8A*). Each black label represents the mean shape score of the listed mutant genotype. The shaded ellipse with an orange border displays the 95% data ellipse of PCs 1 and 2 of DO cranial shape variation. The DO mean shape is contrasted by the mutant variation along PC1. The first PC describes vault size relative to the length of the face. The phenotype shown along the X axis of *Figure 8A* depicts the maximum positive PC1 shape, while the heatmap drawn on the crania represents the local deforma-tions towards the minimum PC1 shape. The positive direction of PC2 describes coordinated variation

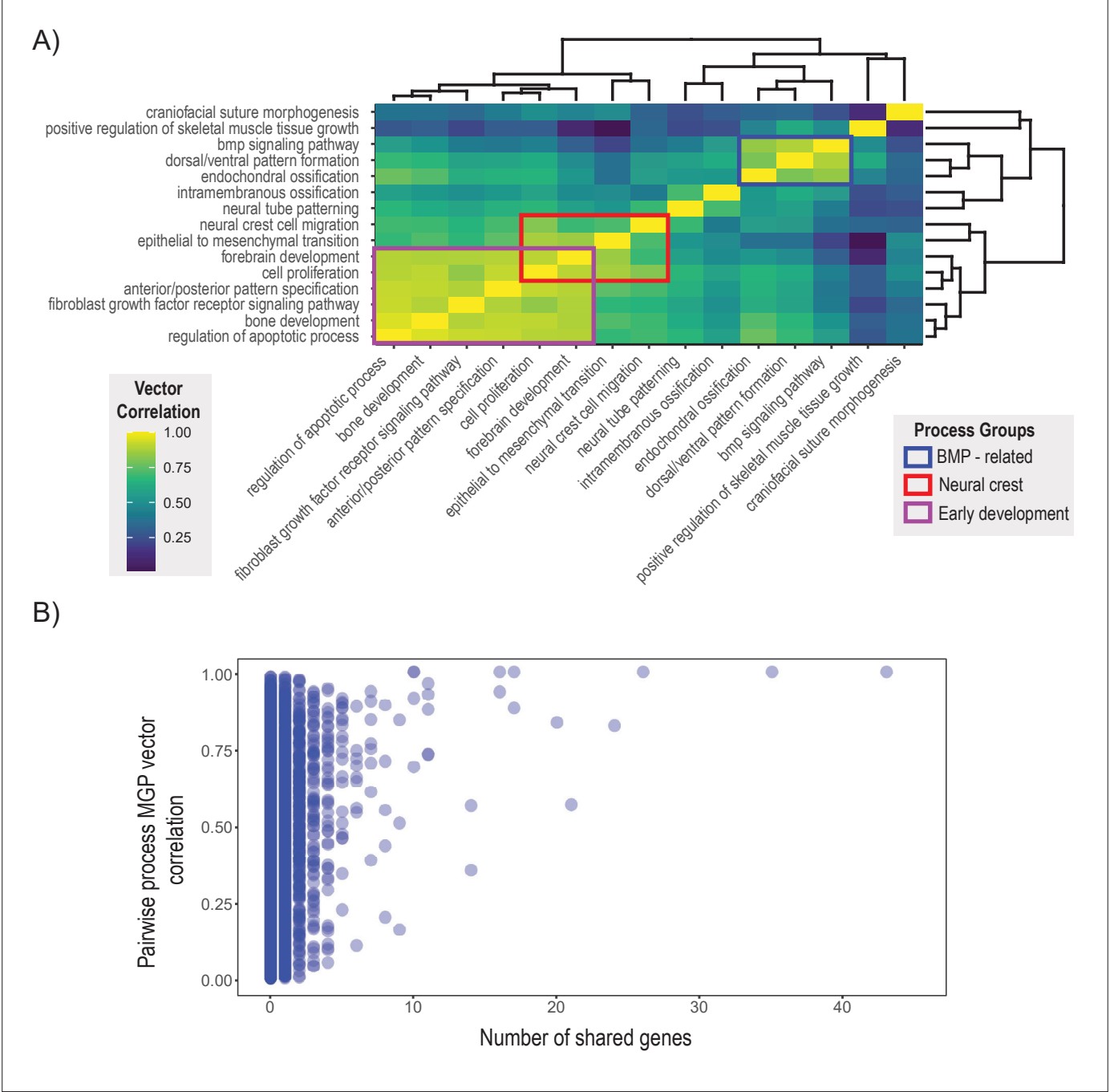

**Figure 7.** Pairwise multivariate genotype-phenotype (MGP) vector correlations. (**A**) Pairwise correlations of phenotypic effects for 15 process MGP analyses. Scale on the right denotes color correspondences to vector correlation, where yellows are high correlations, greens are moderate, and blues are low. (**B**) Pairwise process MGP vector correlations as a function of the number of shared genes between the processes. Processes that share less than 10 genes can produce very similar and very disparate phenotypic effects. Processes with substantial numbers of shared genes will tend to show highly correlated responses as they increasingly use similar marker sets.

The online version of this article includes the following figure supplement(s) for figure 7:

**Figure supplement 1.** Pairwise process multivariate genotype-phenotype (MGP) vector correlations as a function of the number of shared genes between the processes.

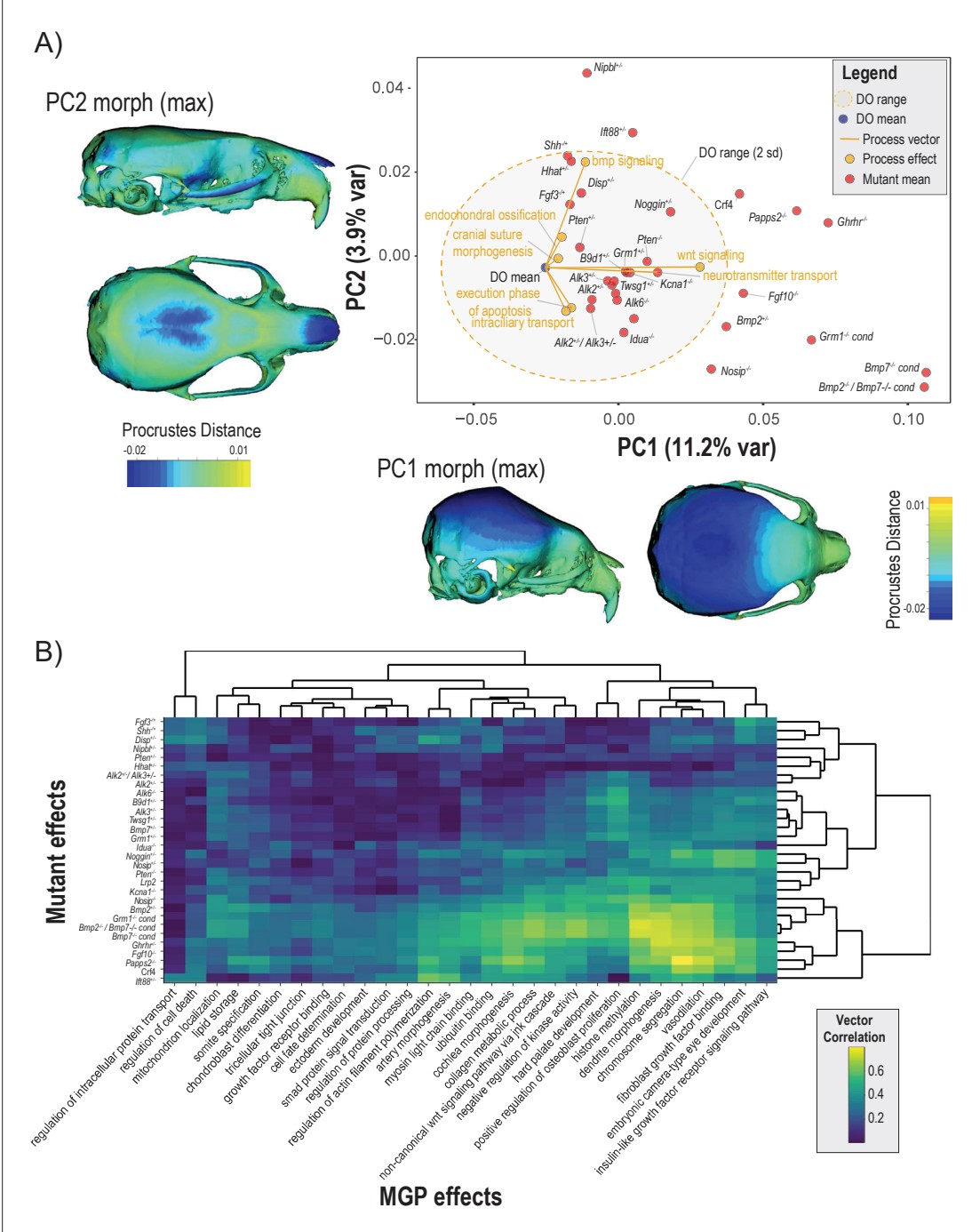

**Figure 8.** Comparisons of multivariate genotype-phenotype (MGP) and mouse mutant directions. (**A**) Seven MGP phenotypes projected onto a principal component analysis (PCA) of the Diversity Outbred (DO) and a sample of 30 mutant mouse genotypes. Mutant means are labeled in black. The directions of MGP effects are shown with orange vectors from the DO mean to the associated process MGP. The range of DO variation on principal components (PCs) 1 and 2 is shown with the shaded ellipse with an orange border. (**B**) A heatmap of vector correlations between 30 mutant effects and 30 process MGP effects. The scale on the right denotes color correspondences to vector correlation, where yellows are high correlations, greens are moderate, and blues are low.

that includes a relatively wider vault, narrower zygomatic, and shorter premaxilla (*Figure 8A*, Y axis margin).

Process effects – highlighted with orange vectors originating at the DO mean shape – are necessarily of smaller magnitude than the total variation in the DO sample. Therefore, to better compare

the direction of process effects the vector magnitudes were magnified 4×. Several process effects align in distinct directions of mutant effects, such as bmp signaling pathway and endochondral ossification in the direction of *Shh, Nipbl,* and *Ift88* mutants. Neurotransmitter transport and Wnt signaling pathway is similar in direction to *Kcna1^Mceph* and *B9d1* mutant effects. Execution phase of apoptosis and intracellular transport both show similar effects to a cluster of Bmp mutants. *Figure 8A* focuses on two PCs, which allows for the contextualization of how process MGP analyses and mutants vary in similar directions and allows us to visualize what those phenotypes look like. This combination of context and visualization is only possible in limited axes and cannot account for differences or similarities in the full multivariate shape space.

To show the similarity of process MGP directions with mutants in the full shape space, we present a heatmap of 30 process MGP effects to 30 mouse mutant models in *Figure 8B*. The heatmap shows the correlation in direction with yellow/green denoting higher correlation and teal/blue denoting lower correlation. The bottom right of heatmap (highlighted by a white border) shows a block of mutants for which there are strong process correlations. These are among the most extreme phenotypes along PC1 (*Figure 8A*) and include mutants for *Nosip, Bmp2, Grm1, Bmp2; Bmp7* transheterozygote, *Bmp7, Ghrhr, Fgf10,* and *Papps2*. The processes most strongly correlated to these mutants are histone methylation, dendrite morphogenesis, chromosome segmentation, vasodilation, and fibroblast growth factor binding.

There are a set of mutant phenotypes that have generally low correlations to the set of processes chosen. These mutants include *Fgf3, Shh, Nipbl, Disp, Pten, Hhat,* and *Alk2; Alk3* transheterozygote. Interestingly, this group of mutants varies more along PC2 than PC1 (*Figure 8A*). Notably, regulation of intracellular protein transport and regulation of cell death are strongly uncorrelated with the majority of mutant directions.

## Real-time process GP mapping

Finally, we provide an online tool to visualize process effects and make comparisons to mutant effects in real time. This application is found at https://genopheno.ucalgary.ca/MGP/ and can be used for analyses similar to those described in this paper. When the user selects GO terms, the program searches for genotype markers adjacent to each gene listed and uses the selected markers to fit a regularized PLS model. The result is an estimate of the many-to-many relationship between the selected markers and cranial shape variation. The visual outputs include barplots depicting the relative allele effect sizes for each gene in the process and a 3D plot of the corresponding axis of shape variation. Users can compare the effects of different processes and also compare process effects to mutant effects from a provided database of 30 mutant genotypes.

To illustrate how to use this application, we have provided the graphical user interface used to select the parameters (*Figure 9*). As an example, in the 'Process text' entry field, supply a starting term; we chose 'brain.' The GO database is then filtered, returning a user-selectable subset of biological process ontology annotation terms in the 'Process filter' field. We chose 'forebrain morphogenesis,' which has 11 associated genes. We chose to magnify the process phenotype vectors 4× and compare the effect to a heterozygous *Ift88* mutant. *Ift88* is a core component of the primary cilia, which are responsible for promoting developmental signals involved in many facets of facial development (*Tian et al., 2017*). Further, the plots that are generated are interactive. For example, marker loadings can be highlighted and subset by genes of interest (*Sievert, 2019*). There is further information about using this online tool in the 'About this app' tab.

## Discussion

A key goal in genomics is to create tractable genetic explanations for phenotypic variation. In this study, we used the process MGP approach to model the joint effects of genomic markers on multivariate craniofacial shape. This approach allows us to address the joint contributions of multiple genes that share ontological characteristic such as pathway membership on craniofacial shape as a multivariate trait. Specifically, we chose markers adjacent to genes annotated under a developmental process of interest. We showed three process MGP analyses in depth, each with distinct phenotypic effects. Each of these comparisons highlighted the integrated structure of phenotypic variation in mouse craniofacial shape. We found that while there are processes with distinct and localized effects, genetic

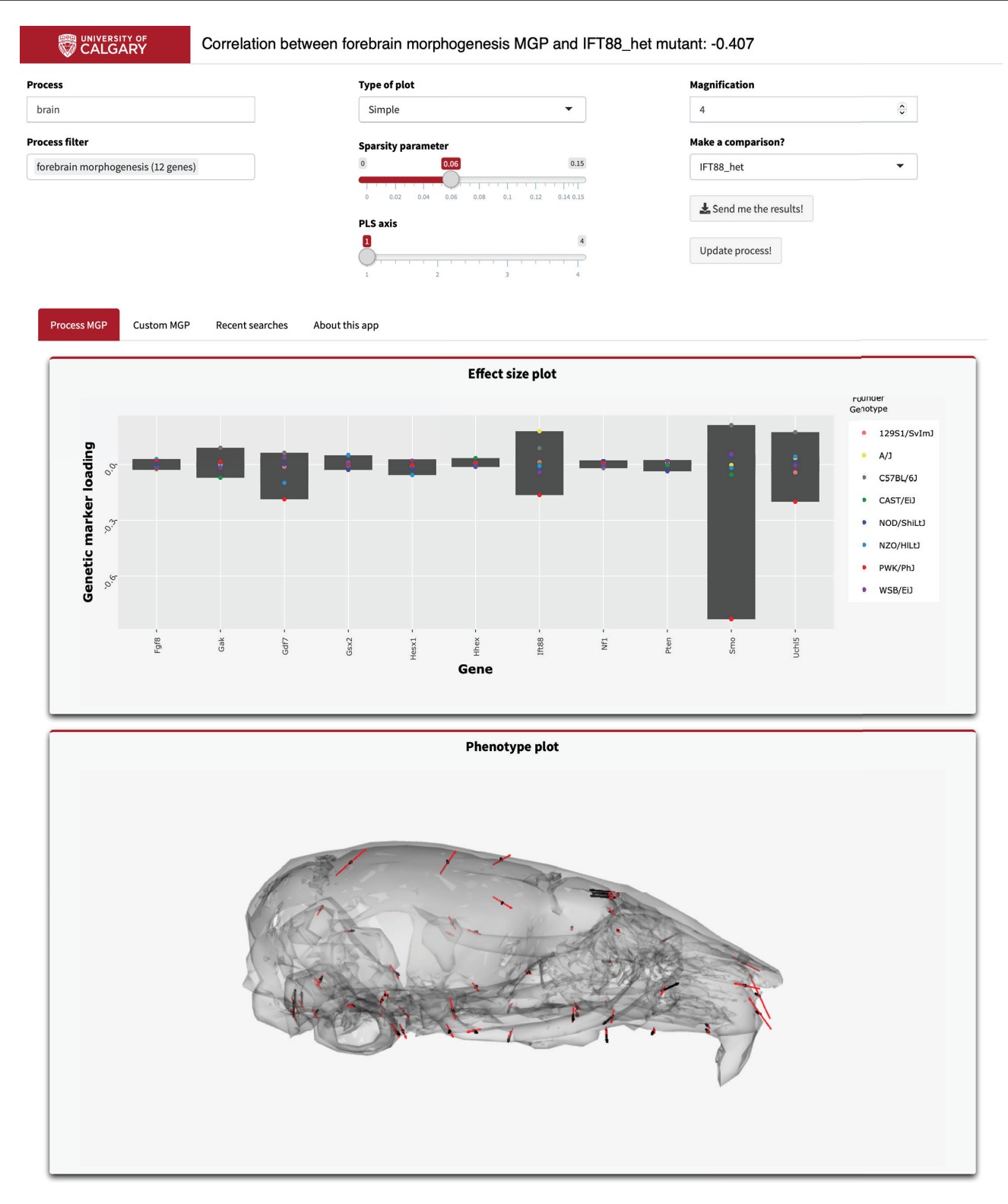

**Figure 9.** Example screenshot of web version of process analysis. Analyses include a barplot of the relative effect sizes of each selected marker and the associated phenotype shown with black vectors at each landmark. If a mutant comparison is selected, the vector correlation is provided and the mutant phenotype is shown with red vectors. Selecting 'send me the results' generates an HTML report with an interactive 3D model.

The online version of this article includes the following figure supplement(s) for figure 9:

**Figure supplement 1.** Combining queries in the multivariate genotype-phenotype (MGP) shiny app with the pipe operator.

effects generally converge on a limited set of directions in phenotype space. Further, these process effects often correspond with the directions of major mutations known to affect these same processes.

Many recent studies have addressed the genetics of craniofacial shape in humans and mice (reviewed in *Roosenboom et al., 2016*; *Weinberg et al., 2018*; *White et al., 2021*). While these studies are yielding a growing list of genes, suggesting that facial shape is highly polygenic, they have left the vast majority of heritable variation unexplained. Existing studies have either used univariate measures of facial shape such as linear measurements or univariate summaries of multivariate shape (e.g., Procrustes distances or PC scores). In addition, most genomic studies of craniofacial shape quantify the effects of each genomic marker independently, with notable exceptions focusing on epistatic effects (e.g., *Varón-González et al., 2019*). Our approach shares common features with some predecessor GP mapping strategies in which candidate genes/SNPs are selected a priori because of common involvement in a pathway (or other mechanistic cluster) (*Claes et al., 2014*; *Liu et al., 2012*; *Wang et al., 2010*; *Wang et al., 2007*). Wang and colleagues selected SNPs based on proximity to genes of interest and effect size to jointly model the pathway-level effects on Parkinson disease data. Their approach is similar to gene set enrichment analysis, weighing overrepresentation of statistical effects related to case-control group membership. In contrast, our approach focuses on estimating a multivariate set of continuous craniofacial responses. Importantly, our approach jointly identifies GP axes that maximally covary. This differs significantly from approaches that determine phenotypes for analysis a priori or based on a predetermined method of data reduction such as the principal component analysis (PCA). Our implementation also differs from similar methods like CCA that was used to associate single-locus effects with a multivariate phenotype (*Claes et al., 2018*). In comparison with the process MGP approach, CCA has the advantage of allowing for a parametric hypothesis test, whereas PLS analyses are limited to permutation-based hypothesis testing. A distinguishing feature of the process MGP approach is the ability to penalize the model with regularization. This is ideal for models with many simultaneous genetic effects in order to mitigate the effects of overfitting. Regularization is not unique to PLS as applications of ridge penalties to CCA have been used for genomic analyses (*Waaijenborg and Zwinderman, 2009*; *Le Floch et al., 2012*).

A key finding of our application of the MGP method to craniofacial shape is that multivariate phenotypic variation aligns nonrandomly to genetic markers associated with pathways or developmental processes. Process MGP effects that are generally not driven by single loci of large effect are possible, like with the chondrocyte MGP analysis (*Figures 2–5A*, *Figure 6—figure supplement 1*). These covarying effects represent the joint genetic effects of multiple contributors to phenotypic variance. While these patterns of MGP covariation may include genetic variants that do not actually affect the phenotype, many other high-loading alleles will be contributors that we lack statistical power to detect under a typical univariate approach (*Pitchers et al., 2019*; *Varón-González et al., 2019*). Here, the overall pattern of GP covariance is the level of genetic explanation for phenotypic variation. When such patterns involve genes that are ontologically linked in meaningful ways, they can provide novel insights into the coordination of genetic effects on phenotypic variation and bolster existing hypotheses from developmental studies.

Another valuable asset that arises from the process MGP approach is the ability to generate testable hypotheses or predictions from MGP observations. The chondrocyte differentiation MGP analysis suggested differentiation defects in the *Bmpr1b* mutant that could contribute to craniofacial variation. We followed up the MGP analysis with histological analysis of *Bmpr1b* mutants and showed premature suture fusion as well as atypical distribution of hypertrophic chondrocytes in the ISS. Similarly, the analysis of left/right symmetry genes suggested that *Fgf10* alleles can contribute to directional asymmetry. A follow-up morphometric analysis of symmetry showed that *Fgf10* mutants do display significant craniofacial asymmetry (*Figure 4D*). Process MGP can also be used to test existing hypotheses about GP relationships. The relative importance of the *Ankrd11* locus in the palate development analysis and the similarity between the genomic and mutant phenotype further validate the role of Ankrd11 in palate development. These examples illustrate the additional insights that a process MGP analysis of a mutational effect can provide. Given that such comparisons can be run quickly with our web application, this creates a tool with the potential for hypothesis generation and initial screening for hypotheses about process-level effects on craniofacial variation in mice.

The explicit modeling of multivariate relationships between phenotypes and genotypes also allows a focus on pleiotropy. Developmental studies in mice demonstrate widespread craniofacial

morphological effects from localized developmental perturbations (*Martínez-Abadías et al., 2012*; *Stelzer et al., 2007*; *Young et al., 2010*) Perturbations to specific processes in development generally produce effects on multiple aspects of phenotype due to knock-on effects at later stages or to interactions at the level of tissues or anatomical structures (*Hallgrímsson et al., 2007*; *Hallgrímsson et al., 2009*). A change in cartilage growth in basicranial synchondroses produces a global change in craniofacial form, for example (*Parsons et al., 2015*). Remarkably, enhancers with highly specific temporospatial effects on gene expression also produce global rather than localized changes in craniofacial shape (*Attanasio et al., 2013*). Given that pleiotropy is likely ubiquitous (*Churchill et al., 2012*; *Wagner et al., 2007*), explicitly multivariate approaches to understanding GP maps are clearly needed.

This convergence of genetic effects on axes of covariation is reflected in our finding that mutations to major developmental genes produce effects that tend to align with the directions of effect associated with the corresponding broader pathways or ontological groups (*Hallgrímsson et al., 2009*). These results suggest that perturbations that are developmentally similar tend to move the phenotype in the same direction in multivariate space (*Figure 8*). Even so, both mutational and higher-level pathway/process effects tend to converge on a few directions of variation, suggesting that multiple pathways and processes lead to common developmental outcomes (*Houle and Fierst, 2013*; *Uller et al., 2018*; *Hallgrímsson and Lieberman, 2008*; *Gonzalez et al., 2013*). This conclusion is further supported by our finding that the genetic axes of covariance for individual processes/pathways can align with multiple directions of mutational effect. For example, the process MGP phenotypes clustered in the bottom right of *Figure 8B* are all highly correlated with a set of BMP and growth hormone-related mutants.

In some cases, mutants and MGP map directions do not correspond. There are several ways this can occur. The first is that the DO population may simply lack alleles as deleterious as found in mutant lines. A small effect allele in the DO may not align with the direction of a mutant almost completely lacking expression of the target gene. Further, there are many examples where a mutation may have different and sometimes even opposite effects depending on genetic background (*Mackay, 2013*; *Percival et al., 2017*). Mutations of major effect may also differ in direction from variants in related genes that have smaller phenotypic effects due to underlying nonlinearities in development (*Green et al., 2017*). Investigating how variants in genes that are functionally related vary in phenotypic effect is an important avenue of inquiry that is revealed by analyses such as those we have performed here. Additionally, relationships between process and mutant effects may stimulate hypotheses about previously unknown or unvalidated interactions between loci or pathways.

A second potential reason that MGP effects may not correspond to major mutation effects is the use of only one PLS axis for each process analysis. With only one axis, we only show the phenotypic direction with greatest covariance with genetic marker variation. If there are multiple large marker effects that do not covary, the weaker marker effect will be masked in the analysis. For instance, there may be a PLS axis for 'chondrocyte differentiation' that corresponds more strongly with the *Bmp2* mutant phenotype. This phenomenon may be particularly prominent for pathways with substantially different mutant effects, like FGF (*Figure 8A*). While we did not delve into the directions outside of the first PLS axis, we have facilitated the selection of lower axes in the web application for users to explore and compare with mutants of interest.

Finally, our analysis shares the limitation of all approaches based on gene annotation data. Incomplete annotation may lead to faulty or incomplete groupings of genes when defining pathway/process hypotheses. Gene annotation is a huge undertaking, and there is substantial variation in the completeness of different process annotations. Many process annotations are manually assigned using inference from the literature, while most are a combination of automated efforts based on transcript similarity and human curation (*Mudge and Harrow, 2015*; *Finger et al., 2017*). Related to this, we assign gene annotation data to genetic markers based on the closest protein-coding region. While this is a reasonable proxy, there will be regulatory sites that affect genes other than the one immediately adjacent and this is a potential source of uncertainty in our analysis (*Forrest et al., 2014*; *Yue et al., 2014*).

In addition, this approach does not currently model the temporal and spatial aspects of gene function throughout development. As a result, alleles of high importance in an MGP analysis do not necessarily produce craniofacial variation through the selected process. A strong allelic effect like *Ccn3* can load heavily in several processes, like 'chondrocyte differentiation,' 'fibroblast migration,'

and 'negative regulation of inflammatory response.' We do not know the mechanism through which any individual allele contributes to variation from an MGP analysis alone. Genomic data with more fine grain measurements of variation in expression and utility of individual loci may be better suited to teasing out the potential mechanisms that alleles produce variation.

The MGP method represents a deliberate decision to trade higher-level insight from GP association data at the expense of statistical certainty about the significance of individual gene effects. The current implementation of the method also does not allow for quantification of individual epistatic effects. Epistasis occurs when the genotypic trait value for a locus is altered by the genotype of a different locus. Such effects generate nonlinear GP maps, but when considered genome-wide, contribute mainly to additive variance (*Cheverud and Routman, 1996*; *Hill, 2017*). The MGP method is additive in that it models only the linear effects of genes. However, since it captures the covariances among genotypic effects, much of this 'additive' variation is likely epistatic in origin.

Complex traits present a massive challenge in genomics because so many are turning out to be enormously polygenic. To generate tractable explanations of the genetic basis for such traits, methods are needed that extract higher-level representation of GP relationships than those that emerge from single-locus-focused approaches. Here, we present a process-driven framework for deriving such higher-level genetic explanations for phenotypic variation. Our approach leverages the biological tendency for developmental processes to produce covariation among aspects of a multi-variate phenotypic trait (*Kawauchi et al., 2009*; *Wagner et al., 2007*). The underlying assumption in this approach is that there are latent variables within high-dimensional GP data that correspond to developmental architecture. We believe that analyses aimed at defining and characterizing such latent variables represent a level of genetic explanation for phenotypic variation that is complementary to genetic analyses designed to establish the significance of single-locus effects. Pursuing such questions will help bridge the gap between emerging mechanistic accounts of morphogenesis and our growing understanding of the genetics of morphological variation.

## Materials and methods
### Mice
We use a sample (n = 1145) of DO mice (Jackson Laboratory, Bar Harbor, ME) to map GP relationships for craniofacial shape (*Churchill et al., 2012*; 2004). The DO is a multiparental outcross population derived from the eight founding lines of the Collaborative Cross (CC). Each animal's genome is a unique mosaic of the genetic diversity found in the CC – more than 45 million segregating SNPs (*Collaborative Cross Consortium, 2012*). Random outcrossing over many DO generations maintains this diversity and, with recombination, increases mapping resolution . Discussions of recommended sample sizes in univariate DO studies can be found in *Churchill et al., 2012*. Both studies recommend a sample size greater than 800 mice for small univariate effect sizes (1–5% variance explained). Further, there are inherent power advantages to our approach because multivariate responses represent maximized differences in phenotype given a set of genotypic measurements. In contrast, univariate approaches such as analyses of individual PCs can only detect effects along those predefined axes that may not have clear biological significance.

Our DO sample was sourced from three separate laboratories and seven DO generations. 386 are from the Jackson Laboratory (JAX), 287 from the University of North Carolina (UNC), and 472 come from the Scripps Research Institute. *Figure 10—figure supplement 1* shows the distribution of the sample by lab source and generation of breeding. Imaging of mice at the University of Calgary was performed under IACUC protocol AC13-0268. Ankrd11 and Bmpr1b mutant mice were bred at the University of Alberta by the Graf lab under Animal Use and Care Committee protocol AUP1149 in accordance with guidelines of the Canadian Council of Animal Care.

### Genotyping
Genotyping was performed by Neogen (Lincoln, NE). Ear clippings were used to extract DNA for all samples. Mice from generations 9, 10, and 15 were genotyped using the MegaMUGA genotyping array (77,808 markers); mice from generations 19, 21, 23, and 27 were genotyped using the larger GigaMUGA array (143,259 markers) (*Morgan et al., 2015*). To pool the genotype data from these two SNP arrays with differing numbers of markers, we imputed markers between the two genotyping

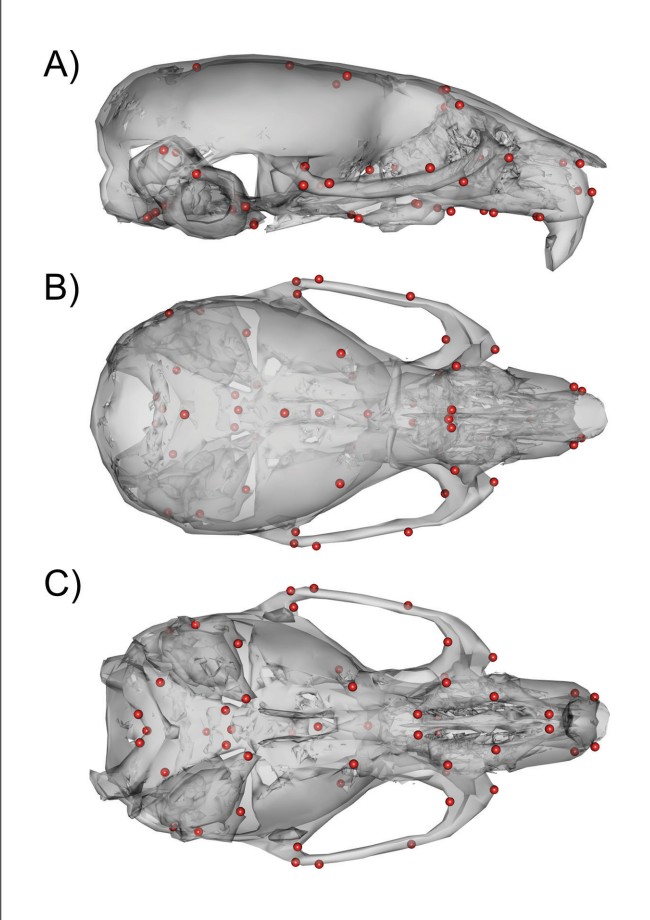

**Figure 10.** 54 3D landmark configuration. (**A**) Sagittal view of representative scan with landmarks shown as red spheres. (**B**) Dorsal view of landmark configuration. (**C**) Ventral view of landmark configuration.

The online version of this article includes the following figure supplement(s) for figure 10:

**Figure supplement 1.** Demographic plots for the Diversity Outbred (DO) sample.

arrays using the 'calc_genoprob' function in the qtl2 package (*Broman et al., 2019*). The function uses a hidden Markov model to estimate genotype probabilities and missing genotype data (*Gatti et al., 2014*). After imputation, the merged genetic dataset consists of 123,309 SNPs that vary among CC founders. Each animal's genetic record is a 123,309 * 8 matrix of estimated diplotype contributions of each CC founder to each marker.

## Scanning and landmarking

We used micro-computed tomography to acquire 3D scans of the full heads of the mice. Scanning was done at the University of Calgary at 0.035 mm voxel resolution (Scanco vivaCT40). One of us (WL) then acquired 54 3D landmarks (*Figure 10*) manually on each volume using Analyze 3D. A discussion of the error associated with manual landmarking can be found in *Katz et al., 2019*. In addition to the DO phenotype data, the mutant mouse data used for comparisons were collected, scanned, and landmarked between the Hallgrimsson and Marcucio labs.

## Landmark registration and analysis

We symmetrized landmarks along the midline of the skull using Klingenberg et al.'s method for object symmetry that configures landmark pairs into a common orientation with reflection and subsequently removes variation associated with translation, scale, and rotation, using Generalized Procrustes Analysis (*Adams et al., 2013*; *Klingenberg et al., 2002*; *Mardia, 2000* et al.; *Schlager, 2017*). We tested for directional asymmetry using the Procrustes ANOVA approach described in *Klingenberg et al.,*

*2002*. To focus on shared, within-generation patterns in our multigenerational DO sample without sex effects, we regressed symmetric shape on DO generation and sex and used the residual shapes with the grand mean added as the observations for analysis.

## Genetic relatedness

Adjustment of phenotypes for the influence of genetic relatedness is a common approach in genomic studies to prevent spurious associations. However, it is not necessary in all cases, such as situations with low genetic relatedness and little variation in relatedness. We evaluated whether accounting for genetic relatedness was important for our sample. To do so, we estimated a kinship matrix based on DO genotype correlations (*Cheng and Palmer, 2013*; *Broman et al., 2019*). The kinship values in our sample have a mean of 0 and a standard deviation of.047. As a result of these findings, we performed all subsequent analyses on the within-generation symmetric shape data, without an adjustment for relatedness.

## Regularized PLS analysis

MGP methods for explicitly modeling multivariate phenotypes and for overcoming the limitations of simple linear regression are increasingly common in mapping studies. One example of a multivariate genomic approach is found in *Claes et al., 2018*, where the authors used CCA to quantify individual SNP effects for a multivariate measurement of facial shape. CCA returns a vector of the linear combination of phenotypic effects that maximally correlates to the alleles at a given locus. *Mitteroecker et al., 2016* developed a similar multivariate strategy around a singular value decomposition (SVD) of GP covariance matrices (versus decomposition of correlation matrices in CCA). PLS describes a family of approaches that use SVD to decompose cross-covariance matrices (*Lee et al., 2011*; *Mitteroecker et al., 2016*; *Singh et al., 2016*). PLS is increasingly used with large genetic datasets in order to model how genomic effects extend to multiple traits (*Bjørnstad et al., 2004*; *Mehmood et al., 2011*; *Tyler et al., 2017*). However, its implementation for MGP mapping is, thus far, much more limited.

SVD decomposes the covariance matrix into three matrices:

$$Y = UDV'$$

where **Y** is the mean-centered covariance matrix, **U** denotes the left singular vectors, a set of vectors of unit length describing the relative weighting of each variable on each axis, and **D** denotes the variance along each axis. **V** denotes the set of right singular vectors. For a full (square, symmetric) covariance matrix, **U** and **V** are identical, and the decomposition is equivalent to PCA. For a non-symmetric matrix of covariances, that is, one describing covariance between two distinct blocks of traits, each successive column of **U** and **V** provides a pair of singular vectors describing the best least-squares approximation of covariance between the two blocks, in order of greatest covariance explained to least.

PLS is most often used to find low-rank linear combinations that maximize covariance between two sets of features. Here, we use the data-driven regularized PLS model implemented in the mddsPLS package to find paired axes that maximize covariance between allelic and shape variation (*Lorenzo et al., 2019*). The model uses a lasso penalty to minimize the coefficients (loadings) towards zero to prevent overfitting (*James et al., 2013*). Overfitting can occur when many genotypic markers are included in the model, particularly when markers are colinear. The genotype block is composed of the full set of DO founder probabilities for each selected marker. Thus, an analysis of 20 markers would estimate 160 genotype coefficients. The phenotype block consists of the full set of 54 3D landmarks (162 phenotype coefficients). In all biological process analyses undertaken herein, we used a regularization parameter of 0.06 and report only the first paired axes of the PLS model, that is, the genotype and phenotype axes that explain the most covariance.

We generate graphical displays of process results using the R packages ggplot2 (Wickham, 2016) and Morpho (*Schlager, 2017*). An example script to reproduce the analyses is provided at https://github.com/J0vid/MGP_shiny/tree/main/analyses (copy archived at swh:1:rev:61fb597c48ced306dd588e289e69c3e3d8f9ce15, *Aponte, 2021*).

## Statistical results and comparisons

We estimate the magnitude and direction of MGP process effects using $R^2$ and vector correlations, respectively. $R^2$ is calculated as the ratio of trace of the predicted model covariance to the trace of the phenotypic covariance matrix. We contextualize the MGP process $R^2$ by comparing it to the $R^2$ value of 10,000 randomly drawn marker sets of the same size. For instance, a process annotated with 40 genes would be compared to 10000 40-gene MGP analyses with random markers selected in each iteration. Random marker selection for permutation is constrained to follow similar patterns of linkage disequilibrium to the observed marker set of interest. The null expectation in this scenario is that gene annotation does not provide better information about coordinated marker effects than a randomly selected set of markers.

Vector correlations between process MGP effects are calculated by taking the Pearson product-moment correlation of the two sets of process PLS1 phenotypic loadings. Vector correlations between process effects and mutant effects are calculated by taking the correlation between the process PLS1 phenotypic loadings and mutant MANOVA coefficients. MANOVA was used to compare the mutant group phenotype against the DO sample specified as the reference group. The coefficients of MANOVA describe the relative weights of each landmark coordinate difference between the DO mean shape and the mutant mean shape.

## Chondrocyte morphometrics

Chondrocyte morphometrics were performed using a novel technique developed by the Marcucio Laboratory. Images of the ISS were stained with H&E, SafO, or picrosirius red and were captured and imported into ImageJ (2–6 sections from at least four mice/genotype/synchondrosis; *Rueden et al., 2017*). Landmarks were placed in a defined order (left, right, top, bottom) of visible chondrocytes in the synchondrosis using ImageJ's multi-tool. Data points were then exported as XY coordinates and imported into Microsoft Excel for calculation of major and minor axes relative to overall width of synchondrosis. Area of individual cells was determined from height and width values based on the assumption that each cell is roughly ellipsoidal. An example of major and minor axis measurements and ellipsoidal area measurements on a slide is provided in *Figure 3—figure supplement 1*.

We compared differences in the distribution of cell sizes along normalized synchondroses between *Bmpr1b* mutants and controls with a mixed effects model approach. We used ellipsoidal area of cell size (in microns) as our dependent variable. For fixed effects, we modeled the normalized synchondrosis position (first and second order), where a value of 0 represents the relative midline of the synchondrosis and values of –1 and 1 represent the most distant cells in that synchondrosis. We also modeled genotype as a fixed effect as well as a genotype by cell position interaction (both first- and second--order interactions). For each individual within each genotype, we measured multiple histological sections. These repeated and nested measurements of cell size in multiple sections for each individual were modeled as random effects. To test for cell size differences between genotypes, we used a likelihood ratio test to compare the full model to a reduced model with the fixed effect of genotype and all genotype interactions removed.

## Visualization tools

We introduce an interactive web application that allows the user to select processes of interest with a graphical user interface; see the resulting craniofacial effect at https://genopheno.ucalgary.ca/MGP/. The web apps were written using the shiny package in R (*Chang et al., 2018*; *R Development Core Team, 2017*). The application dynamically filters the MGI GO database based on the initial user input. Queries will only list GO terms with exact matches. For example, 'chond' will return GO terms that incorporate either 'chondrocyte' and 'mitochondria'.

Multiple queries can be selected. An analysis of 'chondrocyte differentiation' and 'chondrocyte hypertrophy' will select the joint gene set of both processes. Processes with different names can be jointly queried with the pipe operator '|,' which is interpreted as an OR (union) operator. For example, to generate the list of GO terms associated with either apoptosis or WNT, we used the 'apoptosis|WNT' query and selected the processes 'Wnt signaling pathway' and 'execution phase of apoptosis' to perform the analysis on the joint set of associated genes (*Figure 9—figure supplement 1*).

Several other parameters can be specified by the user including the type of plot to be generated for the genetic loadings, the amount of magnification applied to the phenotype effect vectors, the

regularization parameter, and the option to overlay a mutant phenotype for comparison. The comparative database currently includes craniofacial shape contrast data (wild-type vs. mutant) for 30 mutant genotypes. If a mutant comparison is selected, the full set of DO specimens are registered with the mutants added (with size removed). We then provide the vector correlation between the process effect and the mutant effect (see *Figure 9*). The database also includes PC1 of the DO sample for comparison.

The app enables users to save results. A save request will generate and download an HTML report of the analysis that includes several versions of the genetic effect plot and an interactive 3D model of the estimated phenotypic effect. If a mutant comparison is selected, it will also appear in the report.

The application tracks recent searches by the user for their reference. A heatmap of process vector correlations of the PLS phenotype loadings is also available under the 'recent searches' tab. The user can select between a heatmap of the processes in their search history or a random assortment of process correlations from past anonymous user searches.

Finally, we provide programmatic access to our model for both process MGP analyses as well as custom gene lists over the web through an application programming interface (API). Queries can be formatted using curl commands as well as request URLs and return results in JavaScript object notation (JSON) format. Documentation for the available functions and their parameters, as well as examples for queries, can be found at https://genopheno.ucalgary.ca/api/__docs__/. The API was written using the plumber package (*Schloerke and Allen, 2021*) in R, with code available at https://github.com/J0vid/MGP_shiny/tree/main/MGP_API.

## Acknowledgements

*Grants*: NIH-2R01DE019638 to RM, BH, and JC, NSERC 238992-17, CIHR Foundation grant 159920 to BH, CFI grant #36262 to BH and NSERC RGPIN-2014-06311 to DG.

MVG was supported by an Alberta Innovates Postdoctoral Fellowship in Health Innovation.

JDA is supported by an Eyes High fellowship, an Alberta Children's Hospital Research Institute scholarship, and a MITACS graduate fellowship.

## Additional information

### Funding

| Funder | Grant reference number | Author |
| --- | --- | --- |
| National Institutes of Health | 2R01DE019638 | Benedikt Hallgrimsson |
| Natural Sciences and Engineering Research Council of Canada | 238992-17 | Benedikt Hallgrimsson |
| Natural Sciences and Engineering Research Council of Canada | RGPIN-2014-06311 | Benedikt Hallgrimsson |
| Canadian Institutes of Health Research | 159920 | Benedikt Hallgrimsson |

The funders had no role in study design, data collection and interpretation, or the decision to submit the work for publication.

### Author contributions

Jose D Aponte, Conceptualization, Data curation, Funding acquisition, Investigation, Methodology, Project administration, Resources, Software, Supervision, Visualization, Writing – original draft, Writing – review and editing; David C Katz, Conceptualization, Investigation, Methodology, Software, Supervision, Visualization, Writing – original draft, Writing – review and editing; Daniela M Roth, Conceptualization, Investigation, Methodology, Supervision, Visualization, Writing – original draft, Writing – review and editing; Marta Vidal-García, Wei Liu, Investigation, Visualization, Writing

– review and editing; Fernando Andrade, Formal analysis, Investigation, Writing – review and editing; Charles C Roseman, Conceptualization, Formal analysis, Methodology, Supervision, Writing – review and editing; Steven A Murray, Conceptualization, Methodology, Project administration, Resources, Supervision, Writing – review and editing; James Cheverud, Methodology, Project administration, Resources, Supervision, Writing – review and editing; Daniel Graf, Conceptualization, Investigation, Methodology, Supervision, Writing – review and editing; Ralph S Marcucio, Conceptualization, Funding acquisition, Investigation, Project administration, Supervision, Writing – review and editing; Benedikt Hallgrímsson, Conceptualization, Data curation, Funding acquisition, Methodology, Project administration, Resources, Supervision, Writing – original draft, Writing – review and editing

### Author ORCIDs
Jose D Aponte http://orcid.org/0000-0002-1608-8612
Daniela M Roth http://orcid.org/0000-0001-8156-4681
Marta Vidal-García http://orcid.org/0000-0001-7617-7329
Ralph S Marcucio http://orcid.org/0000-0002-0537-818X
Benedikt Hallgrímsson http://orcid.org/0000-0002-7192-9103

### Ethics
The work was performed according to protocols approved and reviewed by animal care committees at the University of Calgary (AC13-0268) and the University of Alberta (AUP1149).

### Decision letter and Author response
Decision letter https://doi.org/10.7554/eLife.68623.sa1
Author response https://doi.org/10.7554/eLife.68623.sa2

## Additional files

### Supplementary files
• Transparent reporting form

### Data availability
All diversity outcross microCT scan and QTL data have been deposited with Facebase (https://doi.org/10.25550/1-731C). Scripts are available at https://github.com/j0vid/MGP_shiny in the analyses folder, (copy archived at https://archive.softwareheritage.org/swh:1:rev:61fb597c48ced306dd588e289e69c3e3d8f9ce15) and the associated online tool is available at https://genopheno.ucalgary.ca/MGP/.

The following previously published datasets were used:

| Author(s) | Year | Dataset title | Dataset URL | Database and Identifier |
|---|---|---|---|---|
| Katz DC, Aponte JD, Liu W, Green RM, Mayeux JM, Pollard KM, Pomp D, Munger SC, Murray SA, Roseman CC, Percival CJ, Cheverud J, Marcucio RS, Hallgrímsson B | 2020 | Facial shape and allometry quantitative trait loci in the Diversity Outbred mouse | https://www.facebase.org/chaise/record/#1/isa:dataset/RID=1-731C | Facebase, FB00001077 |

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
