## [Editor Report]

This paper offers a new take on multivariate genotype-phenotype mapping that identifies the joint phenotypic effect of genes involved in known biological processes that impact craniofacial variation. More specifically, the work expands on the traditional idea of candidate gene investigations into candidate biological process investigations, grouping multiple genes into a single analysis. In doing so, the authors show the joint effects of three strong candidate processes, chondrocyte differentiation, determination of left/right symmetry, and palate development on multidimensional craniofacial shape in the heterogenous Diversity Outbred mouse population.

---

## [Decision Letter]

**Decision letter after peer review:**

Thank you for submitting your article "Shapes and genescapes: Mapping multivariate phenotype-biological process associations for craniofacial shape" for consideration by *eLife*. Your article has been reviewed by 3 peer reviewers, and the evaluation has been overseen by a Reviewing Editor and George Perry as the Senior Editor. The following individuals involved in review of your submission have agreed to reveal their identity: Laura Saba (Reviewer #1); Peter Claes (Reviewer #2).

Essential revisions

All three reviewers emphatically stated they found the discussion and conclusions of this paper to be overstated, and not completely supported by the data. That said, they unanimously felt that a substantial revision of this work could deal with this problem and hence a revision opportunity was suggested over an outright rejection. Overall, the requested revisions do not require new data, but do require some reanalysis of the data. These issues are presented in greater detail in the point by point revisions listed below. In addition, when the reviewers tried to use the web app on the browsers Chrome and Firefox, just when the results appeared, they received an error that it disconnected from the server. Please check the web app stability and where necessary provide guidance on browser use for both Mac and PCs.

1. The use of the word 'process' throughout was not initially clear. It would help if it were clearly defined early on.

2. There is some inconsistency between the results, Figure 2A, and the discussion about how Nov/Ccn3 is annotated.

3. For the likelihood ratio test for differences in the distribution of cell sizes between Bmpr1b mutants and controls, it is important that all the interaction effects that included genotype were eliminated as well.

4. The matric used for judging the stability of the cluster results is often referred to as the cophenetic correlation coefficient. This common terminology may help to clarify the first sentence of the paragraph that starts on line 287. It isn't clear why a t-statistics was used to determine if the correlation coefficient was significantly different from zero. Perhaps a reference is needed.

5. In the 'Process effects in mutant morphospace', the authors point out similarities in the direction of the process vector to some of the mutants. It would be impressive if those process/mutant pairs annotated and if those genes had high weights in the MGP genetic vector. Similar comment for the gene/process pairs in the heatmap.

6. The conclusions starting on line 462 and ending on line 466 were not tested rigorously.

7. More detail about how the mutant MANOVA coefficients were calculated is needed.

8. It is not clear how the regularization parameter was chosen.

9. The paper is presented as the development of a new approach; however, it is not written in that way. The methods section "Regularized PLS analysis" is not accessible for a general public. If the reader is not already familiar with multivariate PLS, they will have no chance of understanding what this paper is all about. The results don't mention the actual genotype-phenotype approach, a description of the process-based MGP mapping should be done first thing in the results. Accordingly, the first figure should be current figure 10. And hopefully the legend will include more details.

10. The approach provides the phenotypic effect of genetic variation in already known pathways but it does not result in new genotype-phenotype associations; this is acknowledged in the text. However, the manuscript suggest that the results generate testable hypothesis which is just not supported by the provided data. The predictions of phenotypic effects are done on genes that are previously known to be involved in pathways that result in such phenotypes. For example, the Bmpr1b gene is annotated in the 'Chondrocyte Differentiation' biological process and genetic variation in such gene is therefore used for the genotype-phenotype analysis, but then, the authors suggest that given the results of the mapping, Bmpr1b should show chondrocyte development phenotypes, which is a circular argument. This example is also true for the other two predictions described in the manuscript. A way of showing that the approach results in testable hypothesis will be to run the genotype-phenotype mapping on pathways that are not previously known to affect craniofacial shape, and then test whether the high loading genes indeed affect the phenotype in the expected way. Until that is shown, the claim that this approach results in new testable hypothesis should be taking with caution and this paper must be revised to reflect this reality.

11. The approach is not new. Multiple other methods perform versions of multivariate genotype-phenotype mapping (this is discussed in the manuscript), but more importantly, the authors state to be using a method previously published by Mitteroecker et al. 2016 with the twist of restricting the analysis to known biological processes. It is not clear in the manuscript how much of their approach is actually new and how much is Mitteroecker's applied to a subset of markers. To highlight some lines where it sounds as if the authors did develop the method are found on lines 390-392 and lines 423-425, but rereading of Mitteroecker et al. 2016 suggests this is not the case. This is a crucial point that is not explained nor discussed at all.

12. One of the challenges with multivariate analyses of this type is how to measure success of the model. In this case, the authors compared their genotype-phenotype results to phenotype results from genetically manipulated mice. As written, it is not clear where the information from the mutant mice came from. Although not explicitly stated, it appears that the mutant data is assigned to a process simply based on the inclusion of the mutated gene in the process being tested. There was no consideration of the genetic loadings of that particular gene in the original MPG model. If the mutated gene is in the process but has very weak genetic loadings, the assumption that the mutant phenotype should be similar to the phenotype associated with the process genetic loading is suspect. Theoretically, the MPG predicted phenotype in the mutated rodent would be more accurate if the mutated gene had a stronger genetic loading. Furthermore, if the mutant data was derived from the literature than it likely includes some publication bias. All genes included likely have some effect on cranial morphology or their results likely would not have been published.

13. Within the manuscript, there is an emphasis on the concordant direction of association between the process MGP axis and the axis of shape variation of a relevant mutant phenotype. However, some ambiguity remains about the relevance of the direction of association between the direction of the process MGP axis and the axis of shape variation of a relevant mutant phenotype. The loadings per gene are presented as negative/positive values in all figures, but it is never explained in the methods what does the sign mean. From the description in the methods, the direction of the correlation between the process PLS1 phenotypic loadings and the mutant MANOVA coefficients cannot be predicted unless you knew that the mutation of the gene was similar to one or more of the 8 founder haplotypes. This would require functional knowledge about the markers/haplotype included in the genetic loadings. Explain this in the methods and figure legends to help the reader actually understand what the PLS is doing.

14. There was also some confusion surrounding the rationale and methodology for estimating the null distribution of the R2 values. More detail is needed on how random marker selection for permutation was constrained to follow similar patterns of linkage disequilibrium to the observed marker set. Also, it is unclear what the rationale for randomly selecting markers rather than randomly selecting genes to generate the null distribution.

15. In general, the section 'Comparison of processes to principal component' lacks statistical rigor and conclusions are supported by the data. It is not relevant to compare all 1000 processes with the principal components without first filtering based on R2 values from the MPG analysis. The overall value of the analysis in this section is unclear.

16. In the discussion, the authors do not make a compelling case for how these types of analyses could be used to generate hypotheses for further studies.

17. As stated in the introduction this is a study of complex relationships between genotypes and phenotypes, and it is important to keep in mind that the technique used here as many others in similar endeavors, remain linear in nature, and therefore limited in exposing potentially non-linear relationships.

18. While the three example processes are interesting and easy to understand or follow in terms of, how this is of interest. I do struggle with the interpretation of the follow-up analyses, including correlating effects of processes, comparing of processes to principal component directions and process effects in mutant morpho-space. It is unclear if all these analyses make sense, and more importantly, the dimensionality into which these explorations are being made is rather low. First, any process effect is modelled by the first PLS-component only. Therefore, is the process and its effect on craniofacial shape modelled completely? Probably not. Second, the morpho-space in the last two mentioned analyses here above, is restricted to the first two principal components only. Therefore, is craniofacial variability modelled completely? Probably not. As a result, Figure 7 A, especially, represents only a sub-dimensional view of process effects versus mutant phenotypes. Discrepancies observed, e.g., none of the Bmp mutants align with the bmp signaling pathway, can be due to the limited linear dimensionality, or because of other underlying genuine biological explanations. However, it is hard to tell.

19. It is suggested by one of the reviewer that a better overlap between PLS and CCA and the use of both in genotype-phenotype associations is welcome. E.g., the discussion simply states, "Our approach also differs from methods that associate single locus effects with a multivariate phenotype (Claes et al., 2018)" without any explanation or insight how and what is different. In fact, the difference is not that great, besides the difference in underlying paradigm (GWAS versus candidate selection) and looking across multiple genes instead, which is the forte of this work. The incorporation of gene-based or haplotype-based MV GWAS into the discussion or introduction is also encouraged.

20. The reviewers need to make a stronger case about the benefit of groups of genes, because of the regularization and the selection of a 1-dimensional latent variable. I.e., It is of interest, for at least the three examples given, to run a gene-by-gene based analysis alongside, and to verify the benefit of the process-based analysis.

21. It is suggest that the authors investigate the effect of the user-defined regularization on their results. Additional cross-validation based results to see how good the first PLS component generalizes is needed. It is recognized that collecting a completely independent replication cohort is outside the scope or typical research resources. However, a more elaborated investigation using cross-validations might help.

22. Somewhat related to the point above, the authors need to provide more information on the dimensionality reductions performed, e.g., please present CV results using multiple PLS components, and provide insights on how many components are expected to be relevant in describing a process and its effect on craniofacial variation. The same applies for the 2-PC morphospace, of interest is to provide insight into choosing only 2PCs and no more?

23. The reviewers found the assessment cluster stability to be circular. Please include some references illustrating this approach or to investigate alternative measures for clustering techniques in machine learning, e.g., the silhouette score.

24. The Comparison of processes to principal component directions, ends with listing processes correlated to PCs. It was hard finding out if these groups listed made sense to be grouped as such. Additional and more explicit information for the reader in those directions is welcome here and in other places where such listing of processes (e.g., the clustering) is done.

25. How are the authors estimating % total phenotypic variance explained by the PLS genotype vector (lines 193, 226)? Is this the R2 they talk about in the methods? If so, please use the same terminology in results and methods sections.

26. The authors show that the %var explained in the first three examples (chondrocytes, palate, and symmetry) is higher than randomly chosen set of genes. But maybe, an additional biologically important point to test is whether pathways that are known to be involved in craniofacial development explain significantly higher phenotypic variance than pathways that are expected not to contribute to such phenotypes. This, besides being a testable hypothesis, will shed light into our understanding of the GO terms annotated to affect craniofacial development, and probably highlight pathways previously not associated with such processes that could be followed up in future work.

27. It is highlighted as a main finding that the phenotypic effect of different biological processes correlate with each other and with single mutants, however it is not discussed how much of that correlation is driven by different processes sharing genes. Could you elaborate on that aspect of the analysis? How many genes are shared and whether that correlates with the correlation of vector directions? I wouldn't be sure about how solid the main finding is unless it is clearly shown that it is not coming from shared genetic effects.

28. Supp Figure 4 shows several covariates for the data, however the methods only described how 'generation' was accounted for. What about sex? Mouse source? I don't have a way of knowing whether you are using them as covariates in your model, or whether you can actually add covariates to the model, because the model is not described in the methods.

29. The Results section is dry and doesn't provide any conclusion for the different sections. This could be improved if the conclusions of the analyses that are currently described in the Discussion section be moved to the relevant Results sections. This will also benefit the discussion that is currently very long and most of it is an actual description of results.

30. The information necessary to understand the figures should be in the corresponding figure legend not in the main text (e.g. l 327-329, l 331-333, l346-349). And, all figures showing results of the MGP analysis will benefit from ranking the genes according to variance of the loadings instead of alphabetical order – so it becomes easier to asses which genes are more important.

[Editors' note: further revisions were suggested prior to acceptance, as described below.]

Thank you for resubmitting your work entitled "Relating Multivariate Shapes to Genescapes Using Phenotype-Biological Process Associations for Craniofacial Shape" for further consideration by *eLife*. Your revised article has been reviewed by 3 peer reviewers and the evaluation has been overseen by George Perry as the Senior Editor, and a Reviewing Editor.

The manuscript has been improved but there are some remaining issues that need to be addressed, as outlined below. The reviewers have discussed their reviews with one another, and the Reviewing Editor has drafted this to help you prepare a revised submission. It is recognized that this is the second request for revisions on this paper.

Essential revisions:

A lengthy discussion was had among the reviewers about clarifying what is meant by new hypothesis. It was agreed that this issue must be addressed, but not does not necessarily need new data or experiments. Two of the three points below directly center on this issue.

1) In line 327-329 (of the tracked changes version) the authors write" "The majority of process MGP analyses demonstrate a similar importance to many alleles, highlighting the main strength process-level analyses over individual marker tests". This is an interesting claim, but what data supports this statement exactly? It would be great if this is supported by some density plot with for example # of genes ranking high (whatever threshold you choose) for each process. The authors have tested ~30 or so processes for some of the analyses. Please clarify if this claim coming from there?

2) The reviews all agreed that there was some level of circularity in the examples shown by the authors. The reviewers strongly felt that the response to comment #10 was not convincing, and the main text was not sufficiently modified to clarify this point. The reviewers agreed that if they remained confused on this point, readers of this manuscript likely would be too. The following is a second attempt at explaining the reviewers point again: all three mutants tested (Bmpr1, Fgf10, and Ankrd11) have previous published data showing their effects on craniofacial shape, sometimes even directly related to the biological process the authors are testing. So, given a) they are functionally annotated to be part of that process, b) there are previous publications -possibly the annotation is derived from those studies, although not necessarily- showing their phenotype effect on craniofacial shape, it does not sound as the authors are generating new hypothesis with their method. This would sound more convincing for the readers if the authors explained better for each example what is their rational for calling each of them a new hypothesis generated by the method. Explicitly list what is previously known or not known about those genes and their effect on craniofacial shape (they reference previous studies, but not in the context of framing how this is different from what they are doing), and be emphatically clear about what is the new "thing" they are trying to show. The reviewers are not saying it is not great to see those genes highly ranked within the process MPG vector, but they are just saying that strong claims of this being a new hypothesis are being made when working with genes for which an effect on craniofacial shape has been shown before. This exact point was also raised in query #16 but there was no new text in the Discussion addressing this point.

3. Related to the point above, there were several comments in by the reviewer previously on how, with the current data, the authors could approach more directly the 'testable new hypotheses' that this method generates, but they were not really implemented. This of course does not diminish the value of the paper, but the opposite would have increased its ability to prove that new hypotheses are indeed generated and proven true (not related to genes obviously related to craniofacial shape). For example, point #26 suggests to-test effects on processes that are not related/or expected to be related to craniofacial shape. The authors reply that is difficult to delineate such processes. The reviewers wondered about considering or at least mentioning 'gonad /sperm / egg development'? Social / mating behavior? Circadian rhythm? Etc.

---

## [Author Response]

Essential revisionsAll three reviewers emphatically stated they found the discussion and conclusions of this paper to be overstated, and not completely supported by the data. That said, they unanimously felt that a substantial revision of this work could deal with this problem and hence a revision opportunity was suggested over an outright rejection. Overall, the requested revisions do not require new data, but do require some reanalysis of the data. These issues are presented in greater detail in the point by point revisions listed below. In addition, when the reviewers tried to use the web app on the browsers Chrome and Firefox, just when the results appeared, they received an error that it disconnected from the server. Please check the web app stability and where necessary provide guidance on browser use for both Mac and PCs.1. The use of the word 'process' throughout was not initially clear. It would help if it were clearly defined early on.

The reviewer is correct. We now realize that this word occurs many times in the paper and is not used consistently. In the majority of contexts, we have used the term to essentially mean “gene annotation” and that has now been corrected this in the revision. We now define process, in the context of gene annotation and that is now corrected in the paper. Where we use “process” we are now clear about the context and whether we are referring to genes linked by membership in a pathway or genes with known associations to a subcellular, cellular or tissue level biological activity.

2. There is some inconsistency between the results, Figure 2A, and the discussion about how Nov/Ccn3 is annotated.

This has been corrected.

3. For the likelihood ratio test for differences in the distribution of cell sizes between Bmpr1b mutants and controls, it is important that all the interaction effects that included genotype were eliminated as well.

The reduced model for the likelihood ratio test did not include interaction terms for genotype and distribution. This has been clarified in the manuscript.

4. The matric used for judging the stability of the cluster results is often referred to as the cophenetic correlation coefficient. This common terminology may help to clarify the first sentence of the paragraph that starts on line 287. It isn't clear why a t-statistics was used to determine if the correlation coefficient was significantly different from zero. Perhaps a reference is needed.

We thank the reviewer for this suggestion and have revised the text to use this term. The reason for the t-test is to determine whether the correlation coefficient is real – that is different from the range one might expect under the null hypothesis of no correlation.

5. In the 'Process effects in mutant morphospace', the authors point out similarities in the direction of the process vector to some of the mutants. It would be impressive if those process/mutant pairs annotated and if those genes had high weights in the MGP genetic vector. Similar comment for the gene/process pairs in the heatmap.

Our results show that one can get directions of change in morphospace that resemble those of mutations of major effect via many different combinations of SNPs. This implies that these directions are determined by developmental processes that can be influenced in multiple ways to produce the same direction of effect. This is, in fact, the central point of the paper. We have clarified the text to make this easier to understand.

6. The conclusions starting on line 462 and ending on line 466 were not tested rigorously.

We agree with the reviewers that testing the significance of process-associated directions in morphospace is currently theoretically intractable. Although we believe there is value in the analysis we presented, we agree that the patterns cannot be tested rigorously. For this reason, we have removed these analyses from the paper. Instead, we have followed the reviewers suggestions to focus more on testing the significance of single vs joint SNP effects on directions of variation. We believe that this change strengthens rather than detracts from the paper.

7. More detail about how the mutant MANOVA coefficients were calculated is needed.

We have provided more detail to clarify this in the statistical results and comparisons section in the methods.

8. It is not clear how the regularization parameter was chosen.

As the reviewer is probably intimating, the choice of regularization parameter represents a tradeoff between the risk of overfitting and the risk of not seeing real patterns in the data. In this case, the choice of parameter was driven by our goal to minimize model error without overly penalizing the phenotypic response in comparison to mutant phenotypes. We have clarified this in the results describing the MGP method and have added a supplemental figure showing a range of regularization parameters for all of the example analyses. The regularization parameter can also be changed in the app for users to experiment with settings and determine the effect on the results.

9. The paper is presented as the development of a new approach; however, it is not written in that way. The methods section "Regularized PLS analysis" is not accessible for a general public. If the reader is not already familiar with multivariate PLS, they will have no chance of understanding what this paper is all about. The results don't mention the actual genotype-phenotype approach, a description of the process-based MGP mapping should be done first thing in the results. Accordingly, the first figure should be current figure 10. And hopefully the legend will include more details.

This is an excellent point. We have moved a general description of the method to the introduction and have moved Figure 10 to Figure 1. Figure 10 is actually placed where it is because of the *eLife* format which has methods coming after results. This is easily fixed and we agree with the reviewer on this.

10. The approach provides the phenotypic effect of genetic variation in already known pathways but it does not result in new genotype-phenotype associations; this is acknowledged in the text. However, the manuscript suggest that the results generate testable hypothesis which is just not supported by the provided data. The predictions of phenotypic effects are done on genes that are previously known to be involved in pathways that result in such phenotypes. For example, the Bmpr1b gene is annotated in the 'Chondrocyte Differentiation' biological process and genetic variation in such gene is therefore used for the genotype-phenotype analysis, but then, the authors suggest that given the results of the mapping, Bmpr1b should show chondrocyte development phenotypes, which is a circular argument. This example is also true for the other two predictions described in the manuscript. A way of showing that the approach results in testable hypothesis will be to run the genotype-phenotype mapping on pathways that are not previously known to affect craniofacial shape, and then test whether the high loading genes indeed affect the phenotype in the expected way. Until that is shown, the claim that this approach results in new testable hypothesis should be taking with caution and this paper must be revised to reflect this reality.

We don’t agree with the logic of this comment. The point here, as we understand it, as that there is circularity or ascertainment bias introduced by the fact that gene annotations are based on known phenotypic effects. The reviewer would be correct in this if the phenotypic outcome in our model was the same thing as the annotation. In other words, the observation that the annotation, “craniofacial shape” reveals a differs in craniofacial shape would be uninteresting and circular. However, for the vast majority of process annotations, the relationship between craniofacial shape and the process represented by that annotation is not known. For the example given, there is no known general association between “chondrocyte proliferation” and craniofacial shape. Most of those annotations would have been derived from studies of cartilage or bone development. It is doubtful that any derive primarily from studies of mouse craniofacial morphology. Thus, we don’t see how the circularity that is stated by the reviewer would actually arise in our data.

11. The approach is not new. Multiple other methods perform versions of multivariate genotype-phenotype mapping (this is discussed in the manuscript), but more importantly, the authors state to be using a method previously published by Mitteroecker et al. 2016 with the twist of restricting the analysis to known biological processes. It is not clear in the manuscript how much of their approach is actually new and how much is Mitteroecker's applied to a subset of markers. To highlight some lines where it sounds as if the authors did develop the method are found on lines 390-392 and lines 423-425, but rereading of Mitteroecker et al. 2016 suggests this is not the case. This is a crucial point that is not explained nor discussed at all.

Mitteroecker explains a framework for many-to-many genomic models. We have taken this framework and applied it to a gene subset strategy. Furthermore, we have introduced regularization to mitigate the overfitting effects commonly observed with models with many parameters. In addition, we have applied this method on a dataset with much higher genotyping SNP resolution and have provided an app that allows users to pursue their own analyses quickly. The combination of these factors are novel, despite previous use of PLS in the genomics literature. We have edited the language in the manuscript to more carefully detail what contributions of this work are novel.

12. One of the challenges with multivariate analyses of this type is how to measure success of the model. In this case, the authors compared their genotype-phenotype results to phenotype results from genetically manipulated mice. As written, it is not clear where the information from the mutant mice came from. Although not explicitly stated, it appears that the mutant data is assigned to a process simply based on the inclusion of the mutated gene in the process being tested. There was no consideration of the genetic loadings of that particular gene in the original MPG model. If the mutated gene is in the process but has very weak genetic loadings, the assumption that the mutant phenotype should be similar to the phenotype associated with the process genetic loading is suspect. Theoretically, the MPG predicted phenotype in the mutated rodent would be more accurate if the mutated gene had a stronger genetic loading. Furthermore, if the mutant data was derived from the literature than it likely includes some publication bias. All genes included likely have some effect on cranial morphology or their results likely would not have been published.

Aside from annotation data, the data presented in this paper are not derived from the literature. The mutants that are presented in this paper and that are available for comparison in the app represent data collection by the Hallgrimsson and Marcucio labs. All of the data were obtained from scans in the Hallgrimsson lab and none of these data were obtained from the literature. We have clarified the origin of the data in the methods section.

13. Within the manuscript, there is an emphasis on the concordant direction of association between the process MGP axis and the axis of shape variation of a relevant mutant phenotype. However, some ambiguity remains about the relevance of the direction of association between the direction of the process MGP axis and the axis of shape variation of a relevant mutant phenotype. The loadings per gene are presented as negative/positive values in all figures, but it is never explained in the methods what does the sign mean. From the description in the methods, the direction of the correlation between the process PLS1 phenotypic loadings and the mutant MANOVA coefficients cannot be predicted unless you knew that the mutation of the gene was similar to one or more of the 8 founder haplotypes. This would require functional knowledge about the markers/haplotype included in the genetic loadings. Explain this in the methods and figure legends to help the reader actually understand what the PLS is doing.

The loadings represent the relative strength of allelic effects in the Diversity Outbred genome. There is no clear interpretation of individual effects of an ordination approach like PLS. However, we have now added an analysis of the effect of the removal of highly-loaded genes in the analysis. This result is now presented in figure 6 of the manuscript.

Except for the rare case of novel mutations that arose during the breeding of the DO mice, all of the SNP variants represent genomic sequences of the original 8 founder strains. It is not clear to us what is meant by similarity to the 8 founder strains. The SNP-phenotype associations are what they are, as is the case in any other genetic study. However, it is in the mutant to MGP comparisons that knowledge of the functional effect is relevant. In those cases, we are testing the similarity between genotype-phenotype associations as determined in the DO data to known function to phenotype associations. Those latter associations are based on studies of those mutants and in one case, we have actually predicted a functional effect from a genotype-phenotype association in the DO data. Beyond this, a SNP by SNP functional analysis of the genomic diversity of the DO mice is obviously intractable.

14. There was also some confusion surrounding the rationale and methodology for estimating the null distribution of the R2 values. More detail is needed on how random marker selection for permutation was constrained to follow similar patterns of linkage disequilibrium to the observed marker set. Also, it is unclear what the rationale for randomly selecting markers rather than randomly selecting genes to generate the null distribution.

We have changed the methodology for estimating the null distribution of R2 values to permute the phenotypes instead. This avoids the disadvantages of random marker selection and attempting to match the original linkage disequilibrium pattern of the observed marker set. These changes have been reflected in the methods as well as the figures.

15. In general, the section 'Comparison of processes to principal component' lacks statistical rigor and conclusions are supported by the data. It is not relevant to compare all 1000 processes with the principal components without first filtering based on R2 values from the MPG analysis. The overall value of the analysis in this section is unclear.

We agree and this section has been removed. See response to point 6.

16. In the discussion, the authors do not make a compelling case for how these types of analyses could be used to generate hypotheses for further studies.

This is an excellent point. We have included a paragraph in the discussion highlighting how MGP analyses can generate hypotheses for further studies.

17. As stated in the introduction this is a study of complex relationships between genotypes and phenotypes, and it is important to keep in mind that the technique used here as many others in similar endeavors, remain linear in nature, and therefore limited in exposing potentially non-linear relationships.

We agree with the reviewers that linearity is a drawback to this approach. The discussion has been expanded to highlight this caveat more clearly. This same criticism could be levied at the majority of genotype-phenotype mapping approaches.

18. While the three example processes are interesting and easy to understand or follow in terms of, how this is of interest. I do struggle with the interpretation of the follow-up analyses, including correlating effects of processes, comparing of processes to principal component directions and process effects in mutant morphospace. It is unclear if all these analyses make sense, and more importantly, the dimensionality into which these explorations are being made is rather low. First, any process effect is modelled by the first PLS-component only. Therefore, is the process and its effect on craniofacial shape modelled completely? Probably not. Second, the morphospace in the last two mentioned analyses here above, is restricted to the first two principal components only. Therefore, is craniofacial variability modelled completely? Probably not. As a result, Figure 7 A, especially, represents only a sub-dimensional view of process effects versus mutant phenotypes. Discrepancies observed, e.g., none of the Bmp mutants align with the bmp signaling pathway, can be due to the limited linear dimensionality, or because of other underlying genuine biological explanations. However, it is hard to tell.

Although the analyses presented in this work may be held to reduced dimensions, they are ultimately driven by the g-p relationships in the data. This is in contrast to a priori reduction of the phenotype dimensionality followed by genomic analysis. We agree that this caveat needs to be made clear to the reader and have added detail to that end, particularly focusing on figure 8 (previously Figure 7).

19. It is suggested by one of the reviewer that a better overlap between PLS and CCA and the use of both in genotype-phenotype associations is welcome. E.g., the discussion simply states, "Our approach also differs from methods that associate single locus effects with a multivariate phenotype (Claes et al., 2018)" without any explanation or insight how and what is different. In fact, the difference is not that great, besides the difference in underlying paradigm (GWAS versus candidate selection) and looking across multiple genes instead, which is the forte of this work. The incorporation of gene-based or haplotype-based MV GWAS into the discussion or introduction is also encouraged.

We thank the reviewers for this suggestion. We have expanded on the similarity between CCA and PLS in the methods as well as the Discussion section. Additionally, we included review of existing gene-based GWAS attempts in the introduction.

20. The reviewers need to make a stronger case about the benefit of groups of genes, because of the regularization and the selection of a 1-dimensional latent variable. I.e., It is of interest, for at least the three examples given, to run a gene-by-gene based analysis alongside, and to verify the benefit of the process-based analysis.

It should be noted that no direct comparison between a many-to-many approach and a gene-by-gene is possible. That is because each gene-by-gene analysis solves for a unique phenotypic effect in multivariate space that must be interpreted individually. From this perspective alone, a many-to-many approach is advantageous because it returns a single set of phenotypic and genotypic responses that can be interpreted in unison. In addition, the many-to-many approach explicitly models the coordination of genetic effects at the process level. One point of entry into such a comparison is the gene drop analysis that we have implemented in the revision. See response to point 6.

21. It is suggest that the authors investigate the effect of the user-defined regularization on their results. Additional cross-validation based results to see how good the first PLS component generalizes is needed. It is recognized that collecting a completely independent replication cohort is outside the scope or typical research resources. However, a more elaborated investigation using cross-validations might help.

We’ve done cross validation for regularization and have included that result in supplemental figure 1. We thank the reviewer for this suggestion.

22. Somewhat related to the point above, the authors need to provide more information on the dimensionality reductions performed, e.g., please present CV results using multiple PLS components, and provide insights on how many components are expected to be relevant in describing a process and its effect on craniofacial variation. The same applies for the 2-PC morphospace, of interest is to provide insight into choosing only 2PCs and no more?

The 2PC figure is meant to visually introduce the concept of the similarity of directions of effects. This has the advantage of interpreting the entire context for those first 2PCs. For example, figure 7A shows “wnt signaling pathway” MGP has a similar direction of effects to many mutants, like Fgf10, and the reader knows the corresponding phenotypic effect using the heatmap figure along the x-axis. In contrast, figure 7B shows compares the full vector correlations for several processes to the same set of mutants, with the caveat that we cannot provide visualizations of what those phenotypes look like. We have clarified the intent of figure 7 and its components in the text.

23. The reviewers found the assessment cluster stability to be circular. Please include some references illustrating this approach or to investigate alternative measures for clustering techniques in machine learning, e.g., the silhouette score.

The reviewers may be over-interpreting the significance of the clusters. We point of the analysis in Figure 7 is that there is a higher-level structure to the vector correlations among processes and that individual processes don’t tend to point in independent directions. We are not interpreting the specific higher level structure here in terms of the number of clusters or their biological underpinnings. As the reviewers intimate, that would be an entirely new level of analysis in which is beyond the scope of the present paper. However, we did pursue the methodological suggestion of the reviewer and calculate a silhouette score. This actually returns a structure of five clusters which is very similar to what we show in Figure 7. Given that the biological nature of these clusters is not a focus of the current paper, however, we have chosen not to include this result in the paper but provide it as Author response image 1. In terms of the reviewer’s suggestion to add references supporting the use of hierarchical clustering, we now provide an appropriate supporting reference (Sneath and Sokal, 1973) and note that this is a widely used method for clustering multivariate data.

**Author response image 1. sa2fig1:** 

24. The Comparison of processes to principal component directions, ends with listing processes correlated to PCs. It was hard finding out if these groups listed made sense to be grouped as such. Additional and more explicit information for the reader in those directions is welcome here and in other places where such listing of processes (e.g., the clustering) is done.

This section has been removed. See point 6.

25. How are the authors estimating % total phenotypic variance explained by the PLS genotype vector (lines 193, 226)? Is this the R2 they talk about in the methods? If so, please use the same terminology in results and methods sections.

The R2 is calculated in the same way as described in the methods. This has been clarified in the text.

26. The authors show that the %var explained in the first three examples (chondrocytes, palate, and symmetry) is higher than randomly chosen set of genes. But maybe, an additional biologically important point to test is whether pathways that are known to be involved in craniofacial development explain significantly higher phenotypic variance than pathways that are expected not to contribute to such phenotypes. This, besides being a testable hypothesis, will shed light into our understanding of the GO terms annotated to affect craniofacial development, and probably highlight pathways previously not associated with such processes that could be followed up in future work.

We thank the reviewers for this suggestion. It is certainly of interest to see if known craniofacial processes affect craniofacial variation in comparison to processes with no obvious relationship to craniofacial development. However, it is difficult to delineate between processes that are known to be involved in craniofacial development vs processes that are not known to be involved in craniofacial development. For example, cell cycle process related gene sets may not directly influence craniofacial shape, but have indirect effect through phenomena like allometry. Virtually any mutation will have differential effects across cell types, and to the extent that the relevant cell types are present in the developing skull and face, they will contribute to craniofacial variation.

27. It is highlighted as a main finding that the phenotypic effect of different biological processes correlate with each other and with single mutants, however it is not discussed how much of that correlation is driven by different processes sharing genes. Could you elaborate on that aspect of the analysis? How many genes are shared and whether that correlates with the correlation of vector directions? I wouldn’t be sure about how solid the main finding is unless it is clearly shown that it is not coming from shared genetic effects.

We thank the reviewers for this suggestion. We have added an analysis of the relationship between the number of genes 2 process MGP analyses share and the similarity of the resultant phenotype. This is presented in figure 7.

28. Supp Figure 4 shows several covariates for the data, however the methods only described how ‘generation’ was accounted for. What about sex? Mouse source? I don’t have a way of knowing whether you are using them as covariates in your model, or whether you can actually add covariates to the model, because the model is not described in the methods.

We thank the reviewers for pointing out these missing details. They have been clarified in the methods section.

29. The Results section is dry and doesn’t provide any conclusion for the different sections. This could be improved if the conclusions of the analyses that are currently described in the Discussion section be moved to the relevant Results sections. This will also benefit the discussion that is currently very long and most of it is an actual description of results.

These are great suggestions and we have moved the summarized descriptions of the analyses to the relevant subsections of the results.

30. The information necessary to understand the figures should be in the corresponding figure legend not in the main text (e.g. l 327-329, l 331-333, l346-349). And, all figures showing results of the MGP analysis will benefit from ranking the genes according to variance of the loadings instead of alphabetical order – so it becomes easier to assess which genes are more important.

We have expanded on figure legends to provide more detail. We have also revised the figures to sort the loadings by variance.

[Editors' note: further revisions were suggested prior to acceptance, as described below.]

Essential revisions:A lengthy discussion was had among the reviewers about clarifying what is meant by new hypothesis. It was agreed that this issue must be addressed, but not does not necessarily need new data or experiments. Two of the three points below directly center on this issue.1) In line 327-329 (of the tracked changes version) the authors write" "The majority of process MGP analyses demonstrate a similar importance to many alleles, highlighting the main strength process-level analyses over individual marker tests". This is an interesting claim, but what data supports this statement exactly? It would be great if this is supported by some density plot with for example # of genes ranking high (whatever threshold you choose) for each process. The authors have tested ~30 or so processes for some of the analyses. Please clarify if this claim coming from there?

We have included a supplemental figure to figure 6 to better support this claim.

2) The reviews all agreed that there was some level of circularity in the examples shown by the authors. The reviewers strongly felt that the response to comment #10 was not convincing, and the main text was not sufficiently modified to clarify this point. The reviewers agreed that if they remained confused on this point, readers of this manuscript likely would be too. The following is a second attempt at explaining the reviewers point again: all three mutants tested (Bmpr1, Fgf10, and Ankrd11) have previous published data showing their effects on craniofacial shape, sometimes even directly related to the biological process the authors are testing. So, given a) they are functionally annotated to be part of that process, b) there are previous publications – possibly the annotation is derived from those studies, although not necessarily – showing their phenotype effect on craniofacial shape, it does not sound as the authors are generating new hypothesis with their method. This would sound more convincing for the readers if the authors explained better for each example what is their rational for calling each of them a new hypothesis generated by the method. Explicitly list what is previously known or not known about those genes and their effect on craniofacial shape (they reference previous studies, but not in the context of framing how this is different from what they are doing), and be emphatically clear about what is the new "thing" they are trying to show. The reviewers are not saying it is not great to see those genes highly ranked within the process MPG vector, but they are just saying that strong claims of this being a new hypothesis are being made when working with genes for which an effect on craniofacial shape has been shown before. This exact point was also raised in query #16 but there was no new text in the Discussion addressing this point.

We thank the reviewers for this point and agree that the flow from MGP analyses to follow up hypothesis testing could be clearer. The paragraphs pertaining to these analyses have been expanded to more clearly lay out (a) what is known about these genes and their involvement in craniofacial development and (b) how each MGP analysis led to follow up hypothesis testing. This has been addressed in the results, as well as the discussion.

For two of the selected genes (Fgf10 and Bmpr1B) background knowledge might suggest that MGP-informed direction of analysis would add little to their functional understanding. As explained below, much of this knowledge is inferred and more importantly the functional involvement of these genes in the chosen processes was not evident from existing literature. As a matter of fact, it had been overlooked for many years. For the third gene (Ankred11) there were only two reports on craniofacial phenotypes in the literature (one is from Daniel Graf who is an author on this paper). MGP informed us to probe for additional, so far overlooked, phenotypes. The demonstration that MGP is able to lead to the discovery of novel phenotypic involvement of ‘well’-studied genes lends strong support to its general applicability as a discovery platform.

Regarding Bmpr1B, we understand the reviewers’ argument but feel that it is actually based on a misunderstanding. In the case of ‘Bmpr1’, we believe that the reviewer is confusing the gene of focus (Bmpr1B) with Bmpr1A. There are well-described craniofacial phenotypes for Bmpr1A (Saito et al. 2012, Hayano et al. 2015, Pan et al. 2017, Liu et al. 2018, Maruyama et al. 2021). In contrast, Bmpr1b is a ‘minor’ Bmp receptor, its expression levels on a tissue level is about two orders of magnitude lower than Bmpr1a (Pan et al. 2017) and to our knowledge there are no reports showing an independent role for Bmpr1b in craniofacial development. There is one paper that describes delayed cranial bone growth in Bmpr1b mutants (CITE) and the study by Yoon et al. 2005 alludes to a craniofacial phenotype, but this is not described in detail nor is its basis analyzed. Both papers are now discussed and cited in the results. Similarly, here is comparatively little known about its role during chondrogenesis. Initial reports described Bmpr1A and Bmpr1B as having overlapping functions (Yoon et al. 2005). More recently, differential requirements have been reported such as the differential requirement of Bmpr1b for hypertrophic chondrocyte differentiation in vitro (Mang et al. 2020). Given this ‘minor’ role, it was surprising to find a strong association of Bmpr1b in the MGP analysis. It is one thing to note that a mouse has an altered craniofacial shape and another entirely to pinpoint the specific cellular basis for that phenotype. Dr. Graf had been working with this particular mutant for years and had never examined the synchondroses until the MGP analysis produced this result. So, this is a clear case of predicting a phenotype that, upon examination, turns out to be there.

A similar point can be made about the Fgf10 mutant. The prediction we made here is that this mutant has altered craniofacial symmetry. This had not previously been reported and there is no annotation for asymmetric skull associated with this mutant. Again, craniofacial shape effects are diverse and so just the fact that it was known that these mice have a craniofacial phenotype would not predict that they would also have directional asymmetry phenotype. It is known that many mutations are associated with elevated fluctuating (random) asymmetry. Directional asymmetry phenotypes, however, are rare and so to observe this after predicting it from the MGP analysis is a pretty clear example of its utility.

Finally, for the Ankrd11 mutant, we are not predicting a craniofacial shape phenotype per se and we agree with the reviewer that this would have been previously known. Rather, we predicted a palatal shape phenotype in Ankrd11 heterozygous mutants based on the MGP analysis. We do agree that there was some circumstantial evidence that might have pointed in this direction in that human KBG syndrome patients do sometimes (20 percent of cases) have palatal abnormalities. Previous and our own work did not report or notice this phenotype, however. We were motivated to look for it based on the MGP analysis. Again, this confirms the strength of the MGP approach.

Based on above explanations, we are really at a loss to understand how the reviewer comes to the conclusion that our analysis is circular, especially for the first two examples. It is worth pointing that a very large proportion of mutations have an effect on craniofacial shape. Facial shape dysmorphism, for example, is reported in 40% of all described human syndromes. But craniofacial shape effects are very diverse and can have myriad possible underlying causes. A reported annotation of “craniofacial” does not make our approach circular because in the vast majority of those cases, the developmental basis of that effect is unknown and the specific nature of that craniofacial phenotype is not quantified or described. We strongly disagree with the contention of the reviewer that readers would come to the conclusion that the MGP approach is circular as we have presented this work multiple times and discussed it with many colleagues and this issue has never come up.

3. Related to the point above, there were several comments in by the reviewer previously on how, with the current data, the authors could approach more directly the 'testable new hypotheses' that this method generates, but they were not really implemented. This of course does not diminish the value of the paper, but the opposite would have increased its ability to prove that new hypotheses are indeed generated and proven true (not related to genes obviously related to craniofacial shape). For example, point #26 suggests to-test effects on processes that are not related/or expected to be related to craniofacial shape. The authors reply that is difficult to delineate such processes. The reviewers wondered about considering or at least mentioning 'gonad /sperm / egg development'? Social / mating behavior? Circadian rhythm? Etc.

As our genetic data consist of a SNP array, any analysis of non-craniofacial processes will still include marker information for genes involved in craniofacial development to the extent that genes are used and reused myriad processes throughout development. For example, if we take the 8 genes annotated to the GO term “spermatid nucleus differentiation” and reverse the search to see what other processes the gene set is associated with, we get back a list of 144 GO terms. Several of these terms would have plausible associations with craniofacial variation, like “negative regulation of apoptotic process” and “developmental growth”. With a SNP array, we lack the resolution to delineate the potential differences in the expression and use of a gene like *pygo2*, that is annotated to both “developmental growth” and “spermatid nucleus differentiation”. This has been made clearer in the discussion via the addition of a new paragraph.